# Chimeric natural products derived from medermycin and the nature-inspired construction of their polycyclic skeletons

Shupeng Yin[1,4], Zhi Liu[1,2,4], Jingjing Shen[1,2], Yuwei Xia[1,2], Weihong Wang[1], Pengyan Gui[1], Qian Jia[1], Konthorn Kachanuban[1,3], Weiming Zhu [1,2] ✉ & Peng Fu [1,2] ✉

Medermycin, produced by *Streptomyces* species, represents a family of antibiotics with significant activity against Gram-positive pathogens. The biosynthesis of this family of natural products has been studied, and new skeletons related to medermycin have rarely been reported until recently. Herein, we report eight chimeric medermycin-type natural products with unusual polycyclic skeletons. The formation of these compounds features some key nonenzymatic steps, which inspired us to construct complex polycyclic skeletons via three efficient one-step reactions under mild conditions. This strategy was further developed to efficiently synthesize analogues for biological activity studies. The synthetic compounds, chimedermycins L and M, and sekgranaticin B, show potent antibacterial activity against *Staphylococcus aureus*, methicillin-resistant *Staphylococcus aureus*, and methicillin-resistant *Staphylococcus epidermidis*. This work paves the way for understanding the nonenzymatic formation of complex natural products and using it to synthesize natural product derivatives.

Antibiotic resistance has become one of the biggest threats to public health[1]. To address this looming problem, antibiotics with new structures and novel mechanisms of action are urgently needed. As an important source of antibiotics, microorganisms have great potential to produce new compounds effective against drug-resistant bacteria[2]. Various microbial natural products have recently attracted much attention from researchers in the areas of biosynthesis and chemical synthesis. Research methods at the gene and enzyme levels, such as genome mining and heterologous expression, are increasingly being used to discover novel natural products and elucidate their biosynthetic pathways. Researchers use genomic sequence information to identify and annotate biosynthetic gene clusters whose gene products are involved in the biosynthesis of novel natural product scaffolds. Most information related to the biosynthesis of secondary metabolites can be traced back to the genome sequences[3–7]. However, the biosynthetic processes of some extremely complex natural products are still unclear, since some involve spontaneous nonenzymatic reactions for their formation. Chimeric natural products comprise a group of structurally novel and diverse compounds that are formed from two or more types of primary or secondary metabolites through well-known reactions such as cycloaddition, condensation, and esterification. The enzymatic catalysis of these reactions has attracted great attention[8–10]. Moreover, chemists have constructed these complex skeletons using efficient synthesis strategies[11–14]. A growing number of natural products possessing unprecedented chimeric frameworks formed through spontaneous chemical reactions have been discovered, such as discoipyrroles[15], pyonitrins[16], and dibohemamines[17]. Discoipyrrole A is formed from three starting metabolites, 4-hydroxysattabacin,

[1]Key Laboratory of Marine Drugs, Ministry of Education of China, School of Medicine and Pharmacy, Ocean University of China, Qingdao 266003, China. [2]Laboratory for Marine Drugs and Bioproducts, Pilot National Laboratory for Marine Science and Technology (Qingdao), Qingdao 266237, China. [3]Department of Fishery Products, Faculty of Fisheries, Kasetsart University, Bangkok 10900, Thailand. [4]These authors contributed equally: Shupeng Yin, Zhi Liu. ✉e-mail: weimingzhu@ouc.edu.cn; fupeng@ouc.edu.cn

anthranilic acid, and 4-hydroxybenzaldehyde, under mild conditions[18]. Aeruginaldehyde and different aminopyrrolnitrin derivatives undergo an abiotic Pictet−Spengler reaction to generate pyonitrins A−D[19]. Dibohemamines A−C are formed via the spontaneous dimerization of bohemamines and formaldehyde[17]. In addition to their fascinating structures, these compounds formed through nonenzymatic reactions also exhibit potent biological activities[15–17].

Mangrove-associated microorganisms, especially actinomycetes, have been demonstrated to be an important source of antibiotics[20]. To discover new compounds with antibacterial activity against drug-resistant pathogens, we developed a library of *Streptomyces* sp. strains isolated from mangrove samples. This library was utilized to screen for antibacterial activity. The results showed that the metabolites of strain OUCMDZ-4982 exhibit significant antibacterial activity against *Staphylococcus aureus* (SA) and methicillin-resistant *Staphylococcus aureus* (MRSA). To obtain the active compounds from this strain, we investigated its secondary metabolites.

Here, we show the discovery of eight chimeric medermycin-type polyketides, chimedermycins A−H (**1–8**), from the strain OUCMDZ-4982. These compounds feature three types of unique polycyclic systems (Fig. 1, types I−III). The nonenzymatic formation of these chimeric skeletons is described, which led us to synthesize analogues **15**, **21–25**, and **27** from medermycin for antibacterial activity screening. Synthetic compounds **23**, **24**, and **27** exhibit potent antibacterial activity against SA, MRSA, and methicillin-resistant *Staphylococcus epidermidis* (MRSE).

## Results

### Isolation and structural elucidation

Strain OUCMDZ-4982 was isolated from a mud sample collected from Hat Chao Mai National Park, Thailand. It was identified as a *Streptomyces* species by analysis of the 16 S rRNA gene sequence. The preliminary study on its major metabolites resulted in the isolation of medermycin, a known antibiotic with strong activity against Gram-positive pathogens[21–23]. Production-scale fermentation of OUCMDZ-4982 was carried out using rice-based solid medium to provide sufficient material for the identification of new metabolites. Normal-phase fractionation of the extract yielded nine fractions. The MS/MS-based molecular networking (Supplementary Fig. 1) indicated that OUCMDZ-4982 can produce higher molecular weight medermycin derivatives. Purification of compounds from the active fraction led to chimedermycins A−H (**1–8**) and medermycin (**9**).

Chimedermycin A (**1**) was determined to have a molecular formula of $C_{40}H_{43}NO_{13}$ based on the high-resolution electrospray ionization mass spectrometry (HRESIMS) peak at *m/z* 746.2798 [M + H]⁺. Analysis of its ¹H and ¹³C NMR data and comparison with those of medermycin (**9**) suggested that compound **1** contains a pyranonaphthoquinone moiety similar to medermycin[21–23]. The other part of compound **1** was determined to be 6-deoxy-dihydrokalafungin (DDHK)[24,25] by analysis of the remaining signals in the ¹H and ¹³C NMR spectra. The connection between the medermycin moiety and DDHK moiety was confirmed by the heteronuclear multiple bond correlations (HMBCs) of H-4 to C-11″, H-5 to C-10″, and H-9″ to C-14 (Supplementary Fig. 2). The relative configurations of chimedermycin A (**1**) were determined by nuclear Overhauser effect spectroscopy (NOESY) correlations (Supplementary Fig. 2), coupling constant analysis (Supplementary Table 1), and ¹³C NMR calculations[26–28] (Supplementary Fig. 3, Supplementary Data 1–4). The absolute configuration of medermycin (**9**) has been confirmed by total synthesis[29]. Moreover, the stereochemistry of the angolosamine ring in the natural medermycin-type derivatives isolated from *Streptomyces* species seems to be constant thus far because the genes related to this angolosamine ring are highly conserved (Fig. 2)[21–23,30–33].

**Fig. 1 | Structures of chimedermycins A–H (1–8).** Naturally occurring compounds with three types of unusual polycyclic skeletons (types I–III).

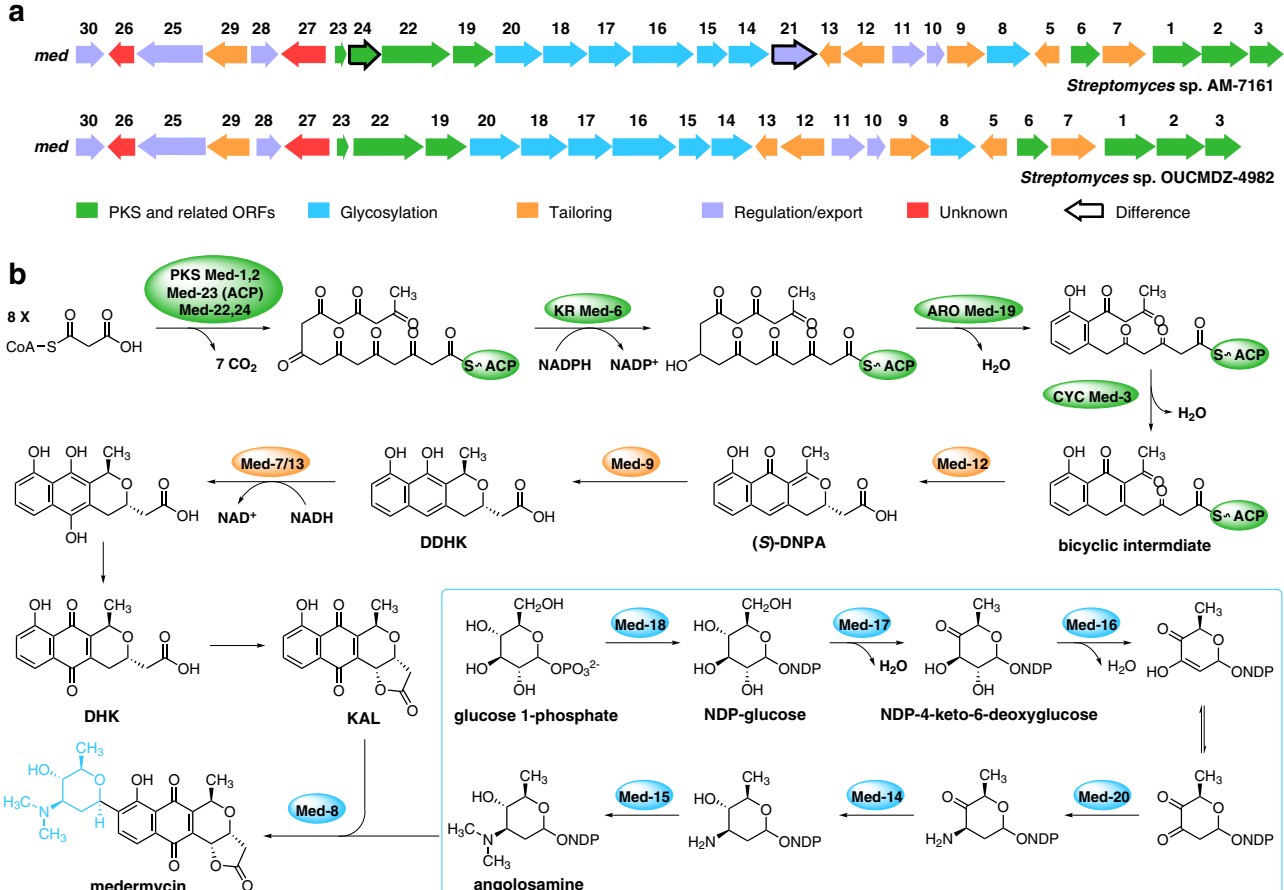

**Fig. 2 | Biosynthetic gene cluster of medermycin and model for its biosynthesis.**
**a** Organization of the *med* cluster in *Streptomyces* sp. AM-7161 and, for comparison, the *med* cluster in *Streptomyces* sp. OUCMDZ-4982. PKS polyketide synthase, ORFs open reading frames. **b** Currently proposed biosynthetic pathway of medermycin. KR ketoreductase, ARO aromatase, CYC cyclase, ACP acyl carrier protein, NADP nicotinamide adenine dinucleotide phosphate, NAD nicotinamide adenine dinucleotide, DDHK 6-deoxy-dihydrokalafungin, DNPA 4-dihydro-9-hydroxy-1-methyl-10-oxo-3-*H*-naphtho-[2,3-*c*]-pyran-3-acetic acid, DHK dihydrokalafungin, KAL kalafungin, NDP nucleoside diphosphate.

Therefore, the absolute configuration of compound **1** was speculated based on the shared biosynthetic origin with medermycin (**9**) (Figs. 2, 3a). This deduction was verified by electronic circular dichroism (ECD) calculations[34] and the ECD exciton chirality method[35]. The ECD spectrum of **1** showed a negative Cotton effect at 344 nm and a positive Cotton effect at 297 nm, which was indicative of negative chirality between the two chromophores (Supplementary Fig. 3). The structure of compound **1** was further confirmed by its semisynthesis from **9** and **10** (Fig. 3b).

Comparison of the NMR spectra of chimedermycins B–D (**2**–**4**) with those of compound **1** revealed that compounds **2**–**4** were methyl ester derivatives of **1** at two different carboxyl groups (Fig. 1). Their structures were further confirmed by the chemical transformations from **2** to **1**, from **1** to **4**, and from **3** to **4**. Analysis of the 1D and 2D NMR data of chimedermycin E (**5**) revealed that it has a chimeric skeleton derived from medermycin and dehydroxy-GTRI-02[36]. The NOESY correlations (Supplementary Fig. 2) and ECD Cotton effects (Supplementary Fig. 4) indicated that compound **5** has the same configuration as compound **1**. Medermycin (**9**) and dehydroxy-GTRI-02 are thus the biosynthetic precursors of chimedermycin E (**5**).

Chimedermycins F–H (**6**–**8**) also contain the same moiety derived from medermycin (**9**) as compound **1**. Their polycyclic skeletons were determined by NMR data analysis (Supplementary Fig. 2, Supplementary Table 3). Compound **7** is the methyl ester derivative of **6**, which was verified by the chemical transformation from **6** to **7**. The structures of compounds **6** and **8** were further confirmed by semisyntheses from

**9** (Fig. 3b). Full details on the structural assignments of chimedermycins A–H (**1**–**8**) can be found in the Supplementary Information.

**Proposed biosynthesis of medermycin and the chimedermycins**

To investigate the formation of the chimeric skeletons, the whole genome of *Streptomyces* sp. OUCMDZ-4982 was sequenced, and the medermycin biosynthetic gene cluster (*med* cluster) in this strain was compared with the reported cluster from *Streptomyces* sp. AM-7161[37,38]. The *med* cluster in *Streptomyces* sp. OUCMDZ-4982 contains 27 open reading frames (ORFs), which are highly homologous with those of *Streptomyces* sp. AM-7161 (Fig. 2a). The product of the missing gene *med-21* is likely associated with pathway regulation (putative kinase) and *med-24* encodes a putative phosphopantetheinyl transferase[39]. Furthermore, no additional genes involved in the biosynthesis of the chimeric frameworks were found in the *med* cluster. Therefore, we speculated that the formation of compounds **1**–**8** may be driven by nonenzymatic reactions. The biosynthetic pathway for medermycin (**9**) has been thoroughly studied (Fig. 2)[37,39–41]. It consists of polyketide formation and attachment of the sugar moiety. First, the key bicyclic intermediate of octaketide origin forms, which further undergoes ketoreduction, cyclization, and dehydration to form the chiral intermediate 4-dihydro-9-hydroxy-1-methyl-10-oxo-3-*H*-naphtho-[2,3-*c*]-pyran-3-(*S*)-acetic acid ((*S*)-DNPA). Subsequent enoyl reduction of (*S*)-DNPA affords 6-deoxy-dihydrokalafungin (DDHK), which undergoes a series of redox reactions to yield dihydrokalafungin (DHK) and

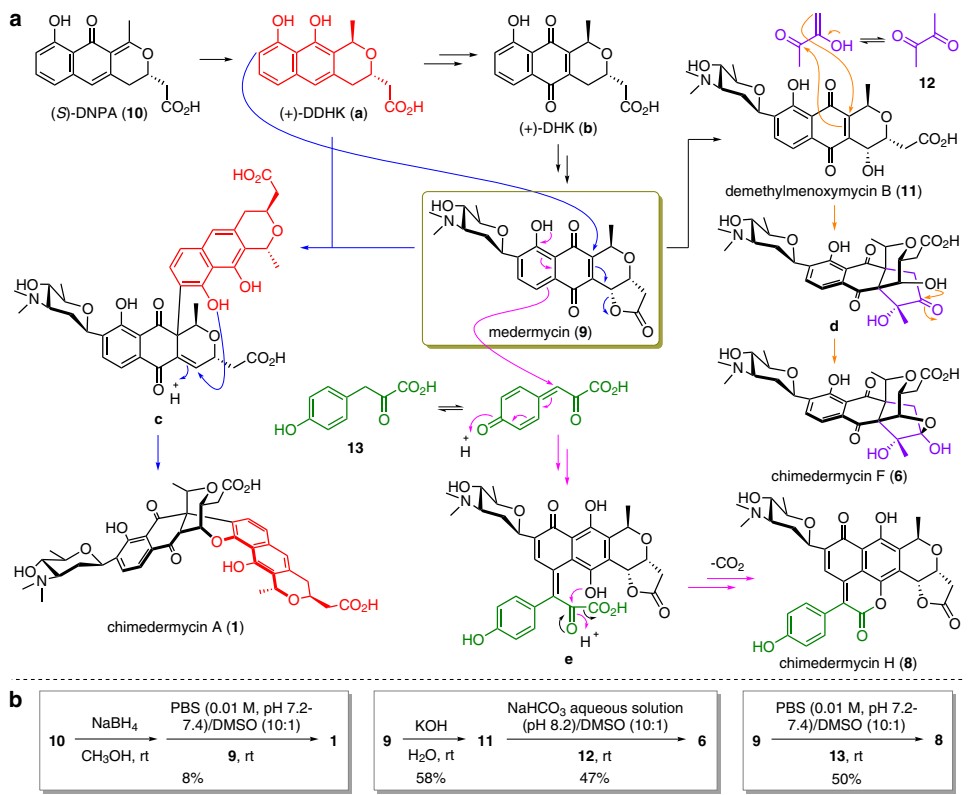

**Fig. 3 | Semisyntheses of chimedermycins A (1), F (6), and H (8). a** Plausible formation mechanism. DNPA 4-dihydro-9-hydroxy-1-methyl-10-oxo-3-*H*-naphtho-[2,3-*c*]-pyran-3-acetic acid, DDHK 6-deoxy-dihydrokalafungin, DHK dihydrokalafungin. **b** Nature-inspired construction of complex polycyclic skeletons from medermycin (**9**). PBS phosphate-buffered saline, DMSO dimethyl sulfoxide.

kalafungin (KAL). Finally, angolosamine, produced via the deoxyhexose pathway, is attached to KAL to generate medermycin (Fig. 2).

Based on the above facts, we proposed a nonenzymatic process for the formation of the chimeric natural products **1–8**. (+)-DDHK, derived from (*S*)-DNPA (**10**), can react with medermycin (**9**) to form chimedermycin A (**1**). It seems likely that (+)-DDHK and medermycin (**9**) undergo Michael addition followed by another nucleophilic addition (Fig. 3a). Medermycin (**9**) is hydrolysed to generate demethylmenoxymycin B (**11**), which can react with butane-2,3-dione (**12**) to form the key intermediate **d** by Michael addition followed by ketol condensation. Then, intermediate **d** undergoes hemiketalization to yield chimedermycin F (**6**) (Fig. 3a). 4-Hydroxyphenylpyruvic acid (**13**) and medermycin (**9**) are considered two biosynthetic precursors of chimedermycin H (**8**). 4-Hydroxyphenylpyruvic acid (**13**) can be easily oxidized into quinone methide by molecular oxygen. Electrophilic attack of quinone methide at the 8-position of medermycin (**9**) followed by dehydro-aromatization results in the formation of intermediate **e**, which undergoes hemiketalization followed by oxidative decarboxylation to generate chimedermycin H (**8**) (Fig. 3a). These small molecules, which react with medermycin to form chimaeras, are derived from different pathways. DDHK is a key intermediate in the biosynthesis of medermycin. Butane-2,3-dione may be produced by *Streptomyces* sp. OUCMDZ-4982. It has been reported that various bacteria can produce butane-2,3-dione. In all these producers, the key enzyme acetohydroxyacid synthase (AHAS; acetolactate synthase) is involved in the biosynthesis of acetolactate from pyruvate. Acetolactate is unstable and can be easily converted to butane-2,3-dione through nonenzymatic decarboxylation in the presence of oxygen[42]. AHAS is a heterotetramer that is composed of two subunits, a catalytic one and a regulatory one, which are encoded by the genes *ilvB* (Gen-Bank accession no. AY785370.1) and *ilvN* (GenBank accession no. AF175526.1), respectively, in a *Streptomyces cinnamonensis* strain[43,44].

We searched these genes in the genome of *Streptomyces* sp. OUCMDZ-4982, and found two genes with high homology to *ilvB* (91% identity) and *ilvN* (90% identity). So, we speculated that *Streptomyces* sp. OUCMDZ-4982 has the ability to produce butane-2,3-dione. 4-Hydroxyphenylpyruvic acid is an important intermediate during the formation of tyrosine in the primary metabolic process[45].

### Semisynthesis of compounds 1, 6, 8 and their analogues

By analyzing the structural characteristics and possible formation pathways of compounds **1**, **6**, and **8**, we think that their skeletons may be synthesized by simple chemical reactions. To investigate the formation of chimedermycin A (**1**), a reaction between medermycin (**9**) and the model compound naphthalene-1,8-diol (**14**) was designed (Fig. 4a). At the outset, inspired by the living environment of microorganisms, this reaction was biomimetically carried out in DMSO/phosphate-buffered saline (PBS; 0.01 M, pH 7.2–7.4) (1:10) at room temperature (rt) (Fig. 4a). Surprisingly, compound **15**, which we named chimedermycin I, formed almost quantitatively within 1 h (Fig. 4b). Furthermore, we investigated the influence of different solvent systems on this reaction (Fig. 4) and found that it hardly occurs in DMSO or DMSO/MeCN (1:10), and only a small amount of product can be observed in DMSO/H₂O (1:10). Although aqueous sodium bicarbonate (pH 7.5) can promote the reaction, this condition is not better than that of the DMSO/PBS system (Fig. 4b).

With the appropriate reaction conditions in hand, we expected to accomplish the semisynthesis of chimedermycin A (**1**) from medermycin (**9**) and (+)-DDHK. Medermycin (**9**) was isolated from the broth of the *Streptomyces* strain OUCMDZ-4982. However, we did not obtain the other substrate (+)-DDHK or its precursor (*S*)-DNPA (**10**) from this strain. We also carefully examined the LC-MS data of the other fractions, but neither target compound was found. Fortunately, in our previous research, we obtained a small amount of (*S*)-DNPA (**10**) from

**a**

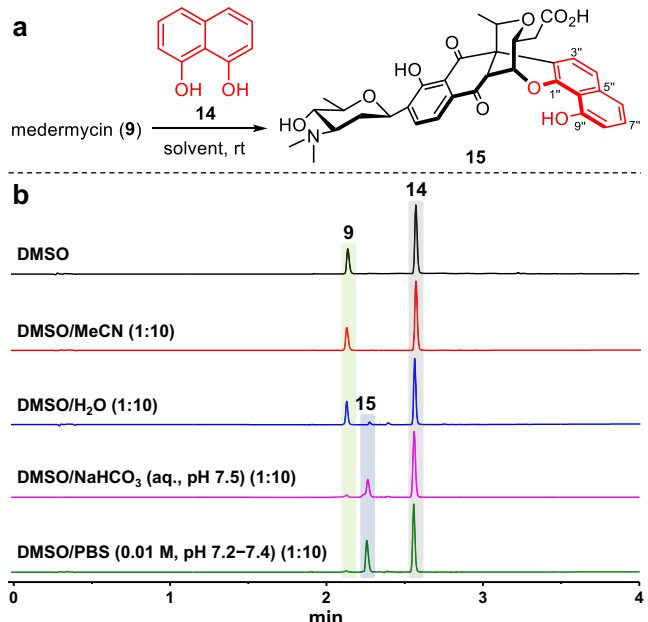

**b**

**Fig. 4 | Synthesis of chimedermycin I (15). a** Reactions of **9** and **14** in different solvent systems to generate **15**. **b** UPLC traces (absorbance measured at 280 nm) of the reaction mixtures after 1 h. PBS phosphate-buffered saline, DMSO dimethyl sulfoxide.

another *Streptomyces* sp. strain (OUCMDZ-4182; GenBank accession no. KU579261) associated with the marine algae *Euphausia superba*[46]. The structure of **10** was identified by comparison of NMR and specific rotation data with reported data[47,48]. Full details on the fermentation of *Streptomyces* sp. OUCMDZ-4182, and the isolation of compound **10** can be found in the Supplementary Information. Compound **10** was treated with $NaBH_4$ to yield (+)-DDHK, which further reacted with medermycin (**9**) in DMSO/PBS (0.01 M, pH 7.2–7.4) (1:10) at rt to generate desired chimedermycin A (**1**) (Fig. 3b).

Inspired by the proposed biosynthetic pathways of chimedermycins F (**6**) and H (**8**) (Fig. 3a), we also attempted to synthesize these two natural products from medermycin (**9**) in a weakly basic aqueous solution. Demethylmenoxymycin B (**11**) is considered a substrate for the synthesis of chimedermycin F (**6**). Lactone hydrolysis of medermycin (**9**) with KOH led to compound **11**, which was identified by spectroscopic analysis (Supplementary Fig. 6). The reaction of **11** and butane-2,3-dione (**12**) was carried out in DMSO/PBS (0.01 M, pH 7.2–7.4) (1:10) at rt, and occurred slowly (after 10 days, substrate remained). To accelerate this reaction, we replaced PBS with $NaHCO_3$ aqueous solution (pH 8.2). This reaction can be completed within 24 h to give chimedermycin F (**6**) (Fig. 3b). Medermycin (**9**) and 4-hydroxyphenylpyruvic acid (**13**) were stirred in DMSO/PBS (0.01 M, pH 7.2–7.4) (1:10) at rt to produce desired chimedermycin H (**8**) in 50% yield (Fig. 3b).

To date, the construction of these three types of natural polycyclic skeletons has been accomplished under mild conditions, which not only confirmed the structures, including the absolute configurations of natural products **1–8**, but also provided effective strategies for the expansion of chemical diversity. Next, we utilized these conditions to synthesize analogues for further biological evaluation. During that process, different substrates, including naphthalene-1,8-diol (**14**), 7-bromonaphthalene-1-ol (**16**), 4-hydroxy-6-methyl-2-pyrone (**17**), 4-hydroxycoumarin (**18**), 2-hydroxy-1,4-naphoquinone (**19**), and pyoluteorin (**20**), were subjected to reaction with medermycin (**9**) to yield chimedermycins I–N (**15** and **21–25**, respectively) (Figs. 4, 5). This efficient reaction can be extended to other polyketides possessing structural features similar to those of medermycin, such as granaticin

A (**26**)[49]. Under the same conditions, the reaction of granaticin A (**26**) and 2-hydroxy-1,4-naphoquinone (**19**) was smoothly carried out, producing compound **27**, which we named sekgranaticin B, in 84% yield (Fig. 6). This reaction proceeded well on the one hundred milligram scale.

## Antimicrobial activity

The antimicrobial activities of compounds **1–9**, **15**, and **21–27** were evaluated against two Gram-negative bacteria (*Pseudomonas aeruginosa* and *Escherichia coli*), three Gram-positive bacteria (SA, MRSA, and MRSE), and two pathogenic fungi (*Candida albicans* and *Candida glabrata*). Given the antibiotic resistance of MRSA and MRSE, ciprofloxacin and vancomycin were chosen as positive controls for antibacterial activity, as these two antibiotics can significantly inhibit methicillin-resistant *Staphylococcus* species. Ketoconazole was used as a positive control for the pathogenic fungi. The antibacterial activity results are shown in Table 1 as minimal inhibitory concentrations (MICs) and minimal bactericidal concentrations (MBCs). These compounds were found to have no activity against Gram-negative bacteria or fungi. Interestingly, the known compounds **9** and **26** exhibited stronger activities against the Gram-positive bacteria SA, MRSA, and MRSE than the positive controls. Compounds **23**, **24**, and **27** showed activities roughly equivalent to vancomycin and slightly stronger than ciprofloxacin (Table 1). To further evaluate the effects of the three new synthetic compounds **23**, **24**, and **27** on bacterial growth, time-dependent killing assays were conducted on MRSA. Within 24 h, the number of bacterial colonies treated with compounds **23** and **24** (at MIC, 2 × MIC, 4 × MIC, and 8 × MIC) remained relatively stable; however, the number of bacteria after treatment with compound **27** (at 2 × MIC, 4 × MIC, and 8 × MIC) was significantly reduced (Fig. 7), indicating that the antibacterial effect of compound **27** is much stronger than that of compounds **23** and **24**. The strategy of combining fermentation and nature-inspired synthesis described in this report can provide enough material for further biological activity studies.

## Discussion

In conclusion, we obtained eight chimeric medermycin-type natural products (**1–8**) from a marine-derived *Streptomyces* species. These compounds contain three types of complex polycyclic skeletons that are formed through spontaneous nonenzymatic reactions. Based on the possible mechanisms of these reactions, we investigated the syntheses of the above mentioned natural products. Astonishingly, these complex structural skeletons can be constructed through three simple one-step reactions, which occur smoothly under near physiological conditions. Using this nature-inspired strategy, we accomplished the semisyntheses of chimedermycins A (**1**), F (**6**), and H (**8**) from medermycin (**9**). Moreover, seven chimeric derivatives, chimedermycins I–N (**15** and **21–25**) and sekgranaticin B (**27**), were efficiently synthesized. During the construction of these chimeric skeletons, some common natural products, such as 4-hydroxycoumarin (**18**) and pyoluteorin (**20**), were involved. This synthetic method provides an effective strategy for the reutilization of simple and common structures and complex and new skeletons for drug development. Three synthetic compounds (**23**, **24**, and **27**) exhibited potent antibacterial activity against Gram-positive bacteria. Hence, obtaining more analogues for further biological studies using this method will be part of our future efforts.

There are a growing number of natural products that are produced by key nonenzymatic steps. Spontaneous reactions, together with enzymes for biosynthesis, play important roles in the formation of complex skeletons. The in-depth investigation of the nonenzymatic formation of natural products can not only help us to further understand the formation mechanism of some complex skeletons but also inspire us to synthesize more compounds for biological activity research. The nonenzymatic formation mechanism of some natural

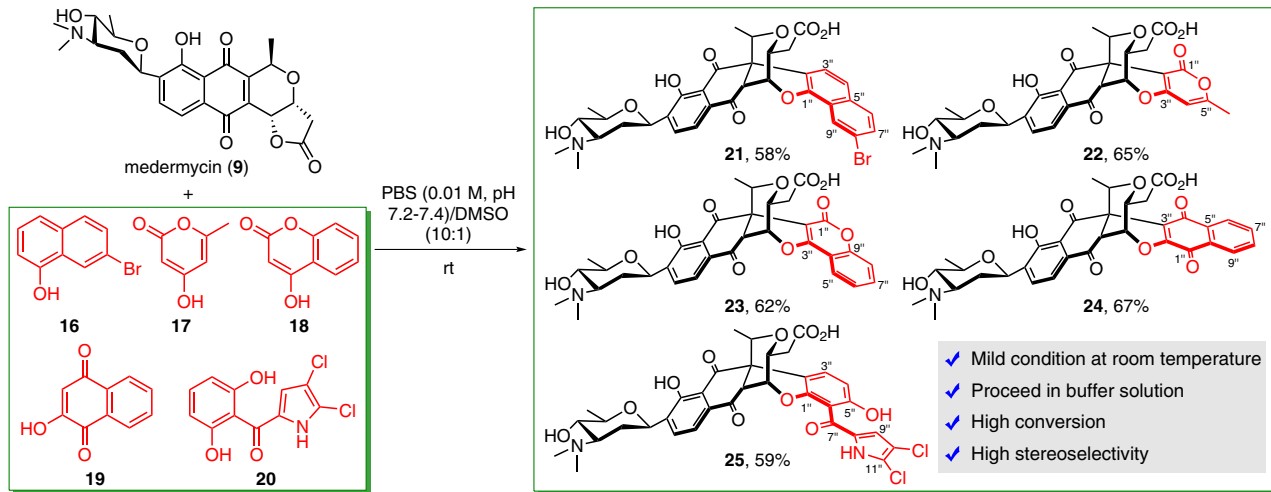

**Fig. 5 | One-step syntheses of chimedermycins J–N (21–25).** The reactions of medermycin (9) and different substrates were carried out in a DMSO/PBS system at rt to yield the desired analogues. PBS phosphate-buffered saline, DMSO dimethyl sulfoxide.

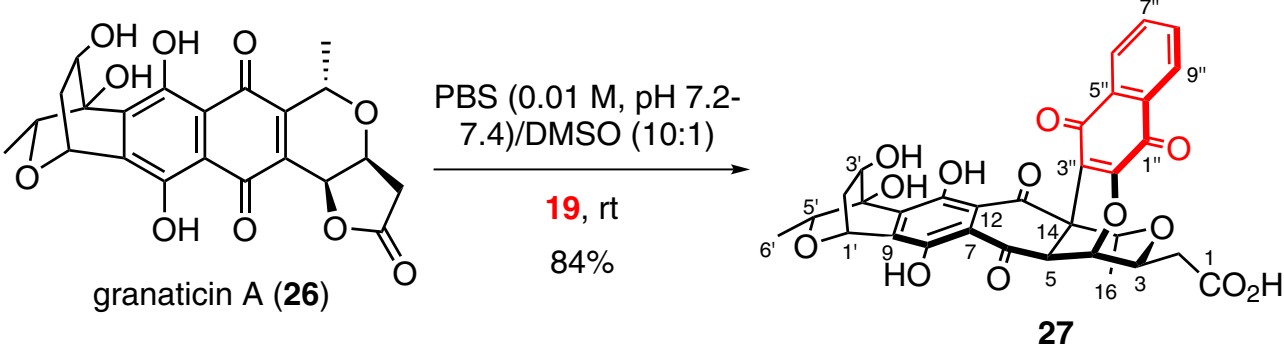

**Fig. 6 | Synthesis of sekgranaticin B (27).** The reaction of granaticin A (26) and 2-hydroxy-1,4-naphoquinone (19) was carried out in a DMSO/PBS system at rt to generate sekgranaticin B (27) on a one hundred milligram scale. PBS phosphate-buffered saline, DMSO dimethyl sulfoxide.

products has been elucidated, but there are still some challenges in this process. Currently, the discovery of nonenzymatic steps remains very accidental. In the future, we need to explore more rational research methods to further understand nonenzymatic biosynthesis, such as utilizing small molecule probes with chemical activity to explore natural products with highly reactive functional groups or structural fragments[50–52]. These compounds may have the potential to form different skeletons with other natural products through spontaneous reactions. On the one hand, it is highly necessary to analyze and verify the formation mechanisms of complex natural products based on chemical principles. On the other hand, analysis of biosynthetic genes can also help us to speculate whether these processes are nonenzymatic. In addition, from an evolutionary perspective, metabolites with high chemical reactivity may have some important ecological functions, but this needs to be confirmed by further systematic study.

## Methods
### General
Optical rotations were recorded with a JASCO P-1020 digital polarimeter. UV spectra were recorded on a 2489 detector of Waters. ECD spectra were measured on JASCO J-815 spectrometer. IR spectra were obtained on a Nicolet Nexus 470 spectrophotometer in KBr discs. NMR spectra were recorded on a Varian System 500 spectrometer, JEOL JNM-ECP 600 spectrometer, Bruker AVANCE NEO 400 spectrometer, or Bruker AVANCE III 600 spectrometer, and chemical shifts were referenced to the corresponding residual solvent signals ($\delta_{H/C}$ 3.31/49.00 for methanol-$d_4$, $\delta_{H/C}$ 2.50/39.52 for DMSO-$d_6$). HRESIMS were measured on a Q-TOF Ultima Global GAA076 LC mass spectrometer. LC-MS data were obtained on a Waters ACQUITY SQD 2 UPLC/MS system with a reversed-phase C18 column (ACQUITY UPLC BEH C18, 2.1 × 50 mm, 1.7 μm) at a flow rate of 0.4 mL/min. Semipreparative HPLC was performed using a C18-PFP column (ACE C18-PFP, 10 × 250 mm, 5 μm), Phenyl column (YMC-Pack Ph, 10 × 250 mm, 5 μm), or π-NAP column (COSIMOSIL π-NAP, 10 × 250 mm, 5 μm). Analytical thin layer chromatography (TLC) was carried out on plates pre-coated with silica gel GF$_{254}$ (10–40 μm). Column chromatography (CC) were performed using silica gel (200–300 mesh, Qingdao Marine Chemical Factory) and Sephadex LH-20 (Amersham Biosciences).

### Collection and phylogenetic analysis of strain OUCMDZ-4982
The actinobacterium *Streptomyces* sp. OUCMDZ-4982 was isolated from a mud sample collected from Hat Chao Mai National Park, Thailand. The sample (2 g) was dried over 24 h in an incubator at 35 °C. The dried sample was diluted to $10^{-3}$ g/mL, 100 μL of which was dispersed across a solid-phase agar plate (Modified Emerson agar media: 1 g of glucose, 1 g of yeast powder, 4 g of peptone, 2.5 g of NaCl, 18 g of agar, and 1 L of seawater) and incubated at 28 °C for 7 days. A single colony was transferred to modified Emerson agar media. Analysis of the 16 S rRNA gene sequence of OUCMDZ-4982 revealed 99.7% identity to *Streptomyces cavourensis*. The gene sequence is deposited in GenBank under accession no. MW193757.

**Table 1 | Antibacterial activity of compounds 1–9, 15, and 21–27 (MIC and MBC values, µg/mL)[a]**

| Compound | SA ATCC 6538 | | | | MRSA ATCC 43300 | | | | MRSE ATCC 35984 | | | |
|---|---|---|---|---|---|---|---|---|---|---|---|---|
| | MIC | | MBC | | MIC | | MBC | | MIC | | MBC | |
| | b | c | b | c | b | c | b | c | b | c | b | c |
| **1** | 32 | 64 | >64 | >64 | 16 | 32 | >64 | >64 | 16 | 64 | >64 | >64 |
| **9** | 0.25 | 1 | 2 | 8 | 0.125 | 0.5 | 4 | 8 | 0.25 | 1 | 1 | 16 |
| **23** | 2 | 2 | 16 | 32 | 1 | 1 | 16 | 16 | 1 | 2 | 16 | 32 |
| **24** | 2 | 4 | 16 | 32 | 1 | 1 | 16 | 16 | 1 | 4 | 16 | 32 |
| **26** | 0.5 | 0.25 | 0.5 | 1 | 0.25 | 0.25 | 4 | 2 | 0.5 | 0.5 | 1 | 8 |
| **27** | 2 | 4 | 2 | 4 | 1 | 2 | 2 | 4 | 1 | 4 | 2 | 8 |
| Ciprofloxacin | 2 | 4 | 4 | 8 | 8 | 8 | 16 | 64 | 4 | 8 | 4 | 16 |
| Vancomycin | 1 | 1 | 1 | 16 | 2 | 2 | 8 | 16 | 2 | 2 | 4 | 32 |

[a]Compounds **2–8**, **15**, **21**, **22**, and **25** did not show antibacterial activity against the three Gram-positive bacteria (*Staphylococcus aureus* ATCC 6538, methicillin-resistant *Staphylococcus aureus* subsp. *aureus* ATCC 43300, methicillin-resistant *Staphylococcus epidermidis* ATCC 35984) (MIC > 64 µg/mL). None of the tested compounds showed activity against the Gram-negative bacteria (*Pseudomonas aeruginosa* ATCC 10145 and *Escherichia coli* ATCC 11775) or fungi tested (*Candida albicans* ATCC 10231 and *Candida glabrata* ATCC 2001) (MIC > 64 µg/mL).
[b]Activity was examined without BSA.
[c]Activity was examined in the presence of 4% BSA.

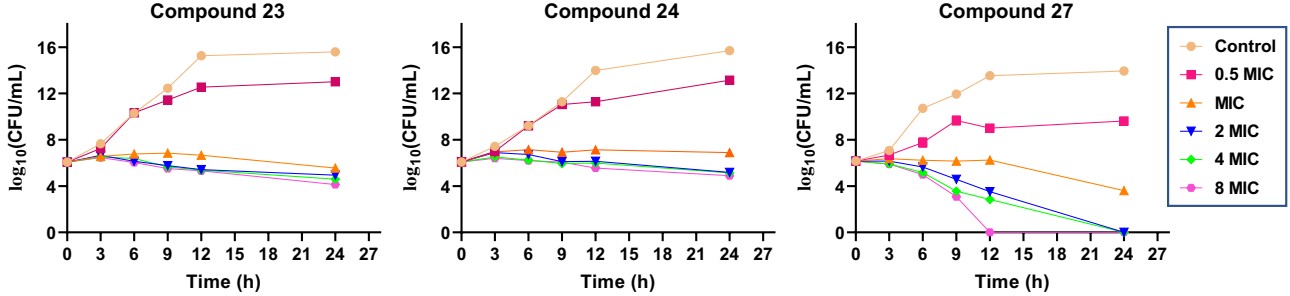

**Fig. 7 | Time-dependent killing of MRSA ATCC 43300 by compounds 23, 24 and 27.** Source data are provided as a Source Data file. MIC minimal inhibitory concentration, CFU colony forming unit.

## Genome sequencing and analysis

The whole genome sequencing of OUCMDZ-4982 was performed using Illumina HiSeq/NovaSeq PE150 platform at Allwegene Technology Company Limited (Beijing, China). All good quality paired reads were assembled using the SPAdes software (v3.13.0) into several scaffolds. The genome was annotated with Prokka software (v1.13.7) and the function of genes was predicted by universal database (NR, SwissProt, GO, KEGG, COG, InterProScan, CAZymes, Pfam, and CARD). The natural product biosynthetic gene clusters of *Streptomyces* sp. OUCMDZ-4982 was analyzed by antiSMASH (v6.0.1) bacterial version (https://antismash.secondarymetabolites.org). The biosynthetic gene cluster of medermycin in *Streptomyces* sp. OUCMDZ-4982 contains 27 ORFs, the function of genes was further analyzed by BlastP in NCBI (https://www.ncbi.nlm.nih.gov) and Uniprot (https://www.uniprot.org). The multiple sequence alignments were performed by DNAMAN.

## Antimicrobial assay

The initial antimicrobial activities against the bacterial and fungal strains (*Pseudomonas aeruginosa* ATCC 10145, *Escherichia coli* ATCC 11775, SA ATCC 6538, MRSA ATCC 43300, MRSE ATCC 35984, *Candida albicans* ATCC 10231, *Candida glabrata* ATCC 2001) were evaluated by an agar diffusion method[53]. The tested strains were cultivated in LB agar plates (5 g of yeast extract, 10 g of tryptone, 5 g of NaCl, 15 g of agar powder, and 1 L of water) for bacteria and in YPD agar plates (10 g of yeast extract, 20 g of peptone, 20 g of glucose, 15 g of agar powder, and 1 L of water) for fungi at 37 °C. Ciprofloxacin lactate and ketoconazole were used as positive controls for bacteria and fungi, respectively. Compounds **1–9**, **15**, **21–27**, and positive controls were dissolved

in MeOH at the concentration of 100 µg/mL. A 10 µL quantity of test solution was absorbed by a paper disk (5 mm diameter) and placed on the assay plates. After 24 h incubation, zones of inhibition were observed. The MICs were determined by the broth microdilution method[54]. Broth dilution used LB growth medium containing increasing concentrations (a twofold dilution series) of the antimicrobial compounds, which was inoculated with $5 \times 10^5$ CFU/mL of pathogenic cells. The MICs were considered to be the lowest concentration that caused total inhibition of bacterial growth. MICs were also determined in triplicate for each strain. Ciprofloxacin lactate & vancomycin and ketoconazole were used as positive controls for bacteria and fungi, respectively. The MBCs were determined by removing 10 µL from each clear well, transferring to fresh media, and incubating for 24 h. The MBC is defined as the lowest test concentration that allows no growth in fresh media. Bovine serum albumin (BSA) was added to LB medium. In the presence of 4% BSA, MIC and MBC values were remeasured.

## Time-dependent killing assay

Time-kill assay was carried out on MRSA ATCC 43300. An overnight culture of cells was diluted 1:1000 in LB medium and adjusted to a final inoculum of $10^5$–$10^6$ CFU/mL in LB medium. Bacteria were then challenged with antibiotics at different concentrations. The concentrations of chimedermycin L (**23**), chimedermycin M (**24**) and sekgranaticin B (**27**) were all set as 0.5 × MIC, MIC, 2 × MIC, 4 × MIC and 8 × MIC. At various time points (0, 3, 6, 9, 12, and 24 h), 50 µL of aliquots were removed and serially diluted ($10$–$10^{12}$-fold dilutions) with sterile LB medium and plated onto LB plates and incubated for 24 h at 37 °C. Colonies were counted and CFU/mL was calculated[55].

## Experimental data and methods

For LC-MS/MS-based molecular networking, see Supplementary Fig. 1. For NMR data, see Supplementary Tables 1–5. For ECD curves, see Supplementary Figs. 4 and 7. For 2D NMR correlations, see Supplementary Figs. 2 and 6. For HRESIMS and NMR spectra, see Supplementary Figs. 8–118. For data of chemical calculations, see Supplementary Tables 6–10 and Supplementary Data 1–4. For experimental procedures, theory and calculation details, detailed structural elucidation, and physical data of compounds, see Supplementary Methods.

## Reporting summary

Further information on research design is available in the Nature Research Reporting Summary linked to this article.

## Data availability

The 16 S rRNA gene sequence of *Streptomyces* sp. OUCMDZ-4982 has been deposited in the GenBank database under accession number of MW193757. The data that support the findings of this study are available within the paper and its Supplementary Information file. Additional data are available from the corresponding author upon request. Source data are provided with this paper.

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

## Acknowledgements

This work was financially supported by the National Natural Science Foundation of China (Nos. 41806086, U1906213, 41876172).

## Author contributions

P.F. and W.Z. conceived and designed the project; P.F. wrote the paper with assistance from S.Y. and Z.L.; S.Y., Z.L., J.S., W.W., P.G., Q.J., and K.K. performed the experiments; Y.X. performed the analysis of biosynthesis. All authors discussed the results and commented on the manuscript. S.Y. and Z.L. contributed equally to this research.

## Competing interests

The authors declare no competing interests.
