## [Peer Review File · Nature Communications]

REVIEWER COMMENTS

Reviewer #1 (Remarks to the Author):

The study "Chimeric Natural Products Derived from Medermycin and Nature Inspired Construction of Their Polycyclic Skeleton" by Yin and co-workers describes the identification of eight chimeric natural products that are composed of a fragment that is derived from the medermycin biosynthetic pathway and the respective other fragment is either a medermycin biosynthetic intermediate or one of two primary metabolites. The structures of the identified metabolites are thoroughly characterized and verified by semisynthesis from the key intermediate medermycin. Semisynthesis of the identified metabolites was achieved using mild, almost physiological conditions which made the authors speculate that the isolated chimeric metabolites are the result of spontaneous reactions between medermycin and the respective other fragment. The semisynthetic strategy was subsequently expanded to generate novel derivatives. All metabolites were subjected to a small panel of antimicrobial bioactivity assays.

I really enjoyed reading the paper. All results are clear, described well and correct conclusions are drawn. In its current form, however, the paper is lacking a story/narrative. What was the goal of the study? The authors start the results section with the structural characterization of the identified metabolites. It is, however, unclear to the reader where the metabolites were isolated from. Why was the bacterium selected? How were the metabolites identified? Based on their bioactivity? Or did the authors screen for medermycin-derived natural products?

The first couple of pages of the results section deal with the in-depth structural characterization of the identified metabolites. I have not gone through all spectra in detail, but it seems to me that the structural characterization was done thoroughly. I think, however, that the level of detail of the structural characterization in the main text is not appropriate for the scope of the journal. While it is crucial to provide this data, I believe that this data should be placed in the supplementary information.

In general, the paper is rather written for a chemical journal like *Org. Lett.* than a Nature subjournal that caters to a broad audience. In my opinion, the manuscript needs some major rewriting to cater to the desired audience. In addition, the language of the paper can be improved both in terms of grammar, style, and the usage of the correct scientific terms. In addition, the correct usage of the tenses needs to be checked throughout the manuscript. Moreover, the authors should go over the manuscript and check for the appropriate wording. Sometimes, certain things are likely the way that the authors state them to be but the wording suggests that it is a fact rather than a likely scenario. In many cases, abbreviations are used but not explained. Especially in the structure elucidation section,

methods that go beyond classical 1 and 2 D NMR experiments have been used and should be introduced to a broader audience.

In its current state, the study is a pure metabolomics study. While the authors purified the fragments that likely undergo spontaneous reactions to yield the purified compounds and showed that they react under mild reaction conditions, it think it would be worthwhile sequencing the genome of the producer to identify the biosynthetic gene cluster. Comparison of the medermycin biosynthetic gene cluster in this strain with the reported gene cluster from the original producer could verify or falsify the current hypothesis that it is non-enzymatic reaction that drives the biosynthesis of the chimeric products. Are there any additional genes in the gene cluster in the strain that the authors were working with or are the genes that are involved in the biosynthesis of the primary metabolites that react with medermycin encoded in the vicinity of the gene cluster or is there a second copy of the gene cassettes responsible for the production of these primary metabolites encoded in the gene cluster? Alternatively, the authors can attempt to identify the chimeric products from the original medermycin producer.

Did the author characterize all medermycin chimeras or are there additional chimeric products produced that were not fully characterized? Molecular networking could answer this question.

In general, I believe that there a multiple paragraphs in the manuscript that could profit from more detailed information. It starts at the beginning with telling the reader why the strain was selected and how the metabolites were identified and extends to sections like the biosynthesis of medermycin especially since key intermediates play an important role in this study. Only abbreviations are used to talk about these intermediates which makes it extremely difficult for the reader to follow.

The discussion section is relatively short and rather serves as a summary section. I believe that the authors are missing a chance here to put their study in a broader scientific context of the more and more frequent discovery of natural products that are produced by key non-enzymatic steps, the challenges and opportunities that come along these spontaneous biosynthetic steps and their possible evolutionary role.

Abstract:

The goal of the study, the problem that this study is tackling, its background and its importance for the scientific community is not clear from reading the abstract. The abstract reads like the abstract of a journal that caters to a community of experts in natural product chemistry rather than a broad scientific community.

Key information is only provided in the abstract but not at the appropriate position in the main text (e.g. there are three types of polycyclic skeletons or that the metabolites were isolated from a *Streptomyces* sp.)

Remove numbers from abstract

Introduction:

Line 32:

What is research on genes and enzymes? Please provide more detail on what kind of research provides insight into biosynthetic pathways. Use and introduce the proper terms like e.g. genome mining.

Line 33:

Replace formation mechanism by biosynthesis

Line 34: replace biosynthetic mechanism by biosynthesis. A biosynthesis usually involves multiple steps and multiple mechanisms.

Line 38:

More precisely these chimeric natural products are formed from either one or more secondary metabolites but also primary metabolites as shown in this paper.

Line 40 ff:

The sentence suggests that chemists were not successful. In addition, these are usually complex synthetic routes and not just single reactions.

Line 55ff:

Where does the library come from? Why were mangrove-associated bacteria selected? Give the readers more information about strain OUCMDZ-4982. From going through the supplementary information, it seems like the strain has already been reported elsewhere. Is there any difference in your strain from the strain with 100% 16 S rRNA sequence identify? Is the strain with 100% 16S rRNA identify a known producer of these kind of compounds?

Line 96: the probability analysis suggests the relative configuration

Line 100: what does absolutely conserved mean in this context?

Line 162ff: It would be worthwhile to briefly describe the proposed model for medermycin biosynthesis here, especially as key intermediates play a pivotal role for this story.

Line 184: reaction conditions were not optimized since the initial conditions were the best, as the authors state in the sentence above.

Line 187: why do the authors think, they were not able to identify the important key intermediate?

Line 189: What is *Euphausia superba*? How was that strain obtained? And why was this strain selected?

Line 193: proposed biosynthetic pathways

Line 226: this sentence is highly speculative. Why are proteins the only likely target?

Line 241 ff: The sentence needs to be rephrased as it is difficult to understand in its current form.

Line 242ff: ... I am not entirely sure what the authors mean with "which provides new ideas for the efficient application of simple natural products". What does application mean here?

The methods section is incomplete e.g. no methods on how the bioactivity assays were performed are provided

Raw data should not be made available at "reasonable request" but should be deposited in the appropriate natural product databases (NMR and MS databases)

Fig 3c. What kind of information can be extracted from this panel since the experimental data differ significantly from the calculated data?

Reviewer #2 (Remarks to the Author):

The manuscript 'Chimeric Natural Products Derived from Medermycin and Nature-Inspired Construction of Their Polycyclic Skeletons' is about eight new chimeric medermycin-type natural products with unusual polycyclic skeletons, and some analogues were synthesized, which the biological activity was also studied and some compounds presented good activity. The skeleton of the compounds presented in the manuscript was never reported and showed more active than the positive control, which showed the compound is not only significant for organic synthesis, but also the potential effects in the drug development. The synthetic process developed will offer an effective method for the development for the future.

Reviewer #3 (Remarks to the Author):

The accelerating emergence of bacterial resistance to current antibiotics, increasing frequencies of multi-resistant and pan-resistant bacterial pathogens (especially superbugs) and the associated failure of antibiotic treatment in bacterial infections represent the most important problems in today's healthcare. This fact consequently results in an urgent need for novel effective antimicrobial compounds. For the above reasons, I consider the submitted manuscript to be very current.

Unfortunately, I cannot recommend the manuscript in its current form for acceptance. Important data and tests are missing in the microbiological part and these need to be supplemented.

Major comments:

1) The methodology for testing antimicrobial activity is incorrectly stated in the supplementary material. It is stated in the text „*Pseudomonas aeruginosa* ATCC10145, *Escherichia coli* ATCC11775, *Staphylococcus aureus* ATCC6538, methicillin-resistant *Staphylococcus aureus* subsp. *aureus* ATCC 43300, methicillin-resistant *Staphylococcus epidermidis* ATCC 35984, *Candida albicans* ATCC 10231, *Candida glabrata* ATCC 2001 were evaluated by an agar dilution method.“ However, the agar

diffusion method is described. The results of this method are not given. How the diffusibility of the test substances in the LB and YPD agars used for detection of inhibition zones was determined? What controls were used for this method?

2) The manufacturer's specification of LB and YPD agars used to determine antimicrobial activity by disk diffusion method is missing.

3) Although the text suggests a possible reason for the bacterial activity of the test substances (it is stated in the text „Compounds 9 and 26 possess high reactivity as Michael acceptors, probably resulting in a rapid reaction with free thiols and amines of some important proteins“), I would expect a more detailed description of the possible mechanism of antimicrobial action. Why was ciprofloxacin chosen as a positive control for antibacterial activity? It is on the basis of the presumed mechanism of action that a control antibiotic with a similar type of effect should be chosen.

4) Why are not given all results of antimicrobial activity, ie also MIC results for tested Gram-negative bacteria and yeasts (*Candida albicans* and *Candida glabrata*). I consider it necessary to supplement these results.

5) I do not consider the testing of antimicrobial activity to be sufficient. Why the type of effect, ie bactericidal or bacteriostatic, has not been determined. Furthermore, the evaluation of bactericidal effect of tested compounds in time (kill-time assay) and determination of antimicrobial activity in the presence of 4% BSA are missing. I consider it necessary to supplement these tests and results.

Minor comments

6) The name *Candida glabrata* is wrong, the correct name is *Candida glabrata*.

7) The term *Streptomyces* strains cannot be used, the correct term is *Streptomyces* species strains, resp. *Streptomyces* sp. strains.

Reviewer #4 (Remarks to the Author):

This well-written manuscript by Zhu, Fu and co-workers concerns a series of eight unusual polycyclic marine (mangrove) bacterial products, the chimermycins A–H. These compounds were identified and characterized through a bioguided approach in the search of novel antibiotic agents against drug-resistant bacteria such as MRSA and MRSE. These compounds are chimeric polyketides in part derived from the known pyranonaphthoquinonic medermycin, chimeric natural products being elaborated from chemical reactions, biocatalyzed or not, between at least two different secondary metabolites.

Besides well-documented descriptions of the structural characterization (including the absolute configuration for several) of these compounds, in particular for the “parent” of this compound series, chimermycin A (1) by full NMR analyses, which were rigorously completed by theoretical calculations, comparisons with closely related natural derivatives, electronic circular dichroism (Cotton effects), hemisynthesis and targeted chemical transformations, the authors identified mild and bioinspired conditions to develop hemisynthesis of chimermycins A, F and H from medermycin in moderate to good yields, as well as a series of analogues.

Comments, questions and suggested corrections:

Fig. 3: I am not totally convinced by the attribution of the absolute stereochemistry of the chimermycin A based on the comparison of the experimental (1) and calculated (1A-D) ECD spectra. Could the authors give additional explanations on their interpretation of these data ?

Page 8: Therefore, the planar part of the structure of chimermycin H (8)...

DNPA: please give full name !

Page 8: ketol in place of aldol !? ketone enolate reacting with a ketone, no aldehyde involved... ketolisation ?

Fig. 4 & pages 8 and 10: How was the 4-hydroxyphenylpyruvic acid (13) oxydized into the methylene quinone ?

And also Fig. 4: formation of e ! Dehydrogenation of C-C linkage and reduction of central quinone ? Please explain !

And from e, after hemiketalization, how is the decarboxylation promoted to generate the lactone moiety of 8 ?

Page 10, second paragraph, line 4: “these conditions”

Page 10: the statement „...the construction of natural product-like compound libraries from medermycin or other polyketides...“ is perhaps a bit too preemptions, since only half a dozen compounds were made, and only one with another polyketide, granaticin A.

Page 11: Antibacterial activity - This section is to my opinion a bit weak, as the quality of the results is not really put forward in terms of perspective and comparison with other antibiotics. Nothing is said about the control used, ciprofloxacin.

The following statement: „Compounds 9 and 26 possess high reactivity as Michael acceptors, probably resulting in rapid reaction with free thiols and amines of some important proteins. Their antibacterial activity as well as toxicity may primarily result from this reason“ sounds speculative, even if plausible chemistry wise, at least in part. Moreover, other compounds from the tested series, although not active,

are also Michael acceptors, so hypotheses are hard to propose on this chemistry feature...

SI: enormous and high quality work done on synthesis and characterization. Proton and carbon NMR spectra are of very good to good quality, however, 2D NMR spectra without any annotation of at least diagnostic correlations is not very useful for the reader.

In conclusion, this work conveys a very interesting and well-documented piece of chemical science on unusual chimeric marine products. The quantity and quality of the data related to the organic synthesis and the structural characterization work are remarkable. Access to natural product-like compound libraries based on the chemistry of these compounds is just here described by some preliminary examples, nonetheless convincing for the future. However, the work on antibacterial activity is weakly presented. For sure, the best part of this manuscript resides in the bioinspired (and simple but efficient) organic synthesis that has been performed on these complex natural products and its possible extension to numerous analogues.

Whether or not this manuscript warrants publication in Nature Communications remains an open question at this stage, but I hope that my comments will help the authors to improve their manuscript in order to meet this objective.

Response to referees

We thank the reviewers for their constructive comments and suggestions, which helped improve the quality of our manuscript. According to these comments and suggestions, we have revised the manuscript very carefully. The changes are highlighted in a yellow background. Comments from separate reviewers are listed below and our replies are in blue.

Reviewer #1 (Remarks to the Author):

The study “Chimeric Natural Products Derived from Medermycin and Nature Inspired Construction of Their Polycyclic Skeleton” by Yin and co-workers describes the identification of eight chimeric natural products that are composed of a fragment that is derived from the medermycin biosynthetic pathway and the respective other fragment is either a medermycin biosynthetic intermediate or one of two primary metabolites. The structures of the identified metabolites are thoroughly characterized and verified by semisynthesis from the key intermediate medermycin. Semisynthesis of the identified metabolites was achieved using mild, almost physiological conditions which made the authors speculate that the isolated chimeric metabolites are the result of spontaneous reactions between medermycin and the respective other fragment. The semisynthetic strategy was subsequently expanded to generate novel derivatives. All metabolites were subjected to a small panel of antimicrobial bioactivity assays.

I really enjoyed reading the paper. All results are clear, described well and correct conclusions are drawn. In its current form, however, the paper is lacking a story/narrative. What was the goal of the study? The authors start the results section with the structural characterization of the identified metabolites. It is, however, unclear to the reader where the metabolites were isolated from. Why was the bacterium selected? How were the metabolites identified? Based on their bioactivity? Or did the authors screen for medermycin-derived natural products?

Thank you very much for the support and suggestions. We reorganized the framework of the manuscript. The goal of the study, the origin of the metabolites, and the information of the bacterium are all introduced in the current version. The isolation guided by the bioactivity and LC-MS and the structure identification are also described.

The first couple of pages of the results section deal with the in-depth structural characterization of the identified metabolites. I have not gone through all spectra in detail, but it seems to me that the structural characterization was done thoroughly. I think, however, that the level of detail of the structural characterization in the main text is not appropriate for the scope of the journal. While it is crucial to provide this data, I believe that this data should be placed in the supplementary information.

We totally agree with you on this issue. So, we moved the full details on the structure assignment to the supplementary information, and only synoptically described the structural characterization in the main text.

In general, the paper is rather written for a chemical journal like Org. Lett. than a Nature subjournal that caters to a broad audience. In my opinion, the manuscript needs some major rewriting to cater to the desired audience. In addition, the language of the paper can be improved

both in terms of grammar, style, and the usage of the correct scientific terms. In addition, the correct usage of the tenses needs to be checked throughout the manuscript. Moreover, the authors should go over the manuscript and check for the appropriate wording. Sometimes, certain things are likely the way that the authors state them to be but the wording suggests that it is a fact rather than a likely scenario. In many cases, abbreviations are used but not explained. Especially in the structure elucidation section, methods that go beyond classical 1 and 2 D NMR experiments have been used and should be introduced to a broader audience.

Thank you for your comments and suggestions. To cater to a broader audience, we made great changes to our writing, especially the description of the NMR experiments. We also checked and revised the grammar and tenses very carefully throughout the manuscript. Some misleading expressions were revised. The abbreviations were explained in the text.

In its current state, the study is a pure metabolomics study. While the authors purified the fragments that likely undergo spontaneous reactions to yield the purified compounds and showed that they react under mild reaction conditions, it think it would be worthwhile sequencing the genome of the producer to identify the biosynthetic gene cluster. Comparison of the medermycin biosynthetic gene cluster in this strain with the reported gene cluster from the original producer could verify or falsify the current hypothesis that it is non-enzymatic reaction that drives the biosynthesis of the chimeric products. Are there any additional genes in the gene cluster in the strain that the authors were working with or are the genes that are involved in the biosynthesis of the primary metabolites that react with medermycin encoded in the vicinity of the gene cluster or is there a second copy of the gene cassettes responsible for the production of these primary metabolites encoded in the gene cluster? Alternatively, the authors can attempt to identify the chimeric products from the original medermycin producer.

Thanks a lot for your constructive comments on this important issue. According to your suggestions, we sequenced the whole genome of *Streptomyces* sp. OUCMDZ-4982 and compared the medermycin biosynthetic gene cluster with the reported one from the original producer *Streptomyces* sp. AM-7161. No additional genes in the gene cluster was found. In addition, no chimeric products was identified from *Streptomyces* sp. AM-7161 and other medermycin producers reported. To a certain extent, these results verified our hypothesis that it is nonenzymatic reaction that drives the biosynthesis of the chimeric products.

Did the author characterize all medermycin chimeras or are there additional chimeric products produced that were not fully characterized? Molecular networking could answer this question.

Thanks for your valuable suggestion. The MS/MS-based molecular networking was used to explore the additional chimeric products that had not been characterized. The analysis indicated that some minor chimeric analogues may not be isolated. However, to identify them requires a very large amount of fermentation.

In general, I believe that there a multiple paragraphs in the manuscript that could profit from more detailed information. It starts at the beginning with telling the reader why the strain was selected and how the metabolites were identified and extends to sections like the biosynthesis of medermycin especially since key intermediates play an important role in this study. Only abbreviations are used to talk about these intermediates which makes it extremely difficult for the reader to follow.

This strain was selected based on its significant antibacterial activity, which was added in the text. Then, the isolation and identification of the metabolites were briefly described. The biosynthesis of medermycin was also embodied. The abbreviations were explained when they first appeared in the text.

The discussion section is relatively short and rather serves as a summary section. I believe that the authors are missing a chance here to put their study in a broader scientific context of the more and more frequent discovery of natural products that are produced by key non-enzymatic steps, the challenges and opportunities that come along these spontaneous biosynthetic steps and their possible evolutionary role.

Thanks a lot for your valuable suggestions. We increased our discussion on nonenzymatic biosynthesis. The challenges and opportunities were analyzed. The prospects of application were also included.

Abstract:

The goal of the study, the problem that this study is tackling, its background and its importance for the scientific community is not clear from reading the abstract. The abstract reads like the abstract of a journal that caters to a community of experts in natural product chemistry rather than a broad scientific community.

Thanks very much for your constructive comments. We rewrote the abstract, which included the goal, the problem tackled, the background, and the importance of this study.

Key information is only provided in the abstract but not at the appropriate position in the main text (e.g. there are three types of polycyclic skeletons or that the metabolites were isolated from a *Streptomyces* sp.)

The key information appearing in the abstract was also added in the main text.

Remove numbers from abstract

The numbers were removed.

Introduction:

Line 32: What is research on genes and enzymes? Please provide more detail on what kind of research provides insight into biosynthetic pathways. Use and introduce the proper terms like e.g. genome mining.

This sentence was rewritten. The research methods at the gene and enzyme level, that provide insight into biosynthetic pathways, were mentioned, such as genome mining and heterologous expression.

Line 33: Replace formation mechanism by biosynthesis

It was revised.

Line 34: replace biosynthetic mechanism by biosynthesis. A biosynthesis usually involves multiple steps and multiple mechanisms.

It was revised.

Line 38: More precisely these chimeric natural products are formed from either one or more

secondary metabolites but also primary metabolites as shown in this paper.

It was changed to “primary or secondary metabolites”.

Line 40 ff: The sentence suggests that chemists were not successful. In addition, these are usually complex synthetic routes and not just single reactions.

This sentence was rewritten. The “reactions” was replaced by “synthesis strategies”.

Line 55ff: Where does the library come from? Why were mangrove-associated bacteria selected? Give the readers more information about strain OUCMDZ-4982. From going through the supplementary information, it seems like the strain has already been reported elsewhere. Is there any difference in your strain from the strain with 100% 16 S rRNA sequence identify? Is the strain with 100% 16S rRNA identify a known producer of these kind of compounds?

In order to discover new compounds with antibacterial activity against drug-resistant pathogens, we developed a library of *Streptomyces* sp. strains isolated from mangrove samples. This library contains about 300 strains. We have been working on discovering active compounds from mangrove-associated actinomycetes. Analysis of the 16S rRNA gene sequence of OUCMDZ-4982 revealed 99.7% identity to *Streptomyces cavourensis*. However, we did not find this kind of natural products from these *Streptomyces cavourensis* strains reported. There is a mistake in our previous manuscript that the identity of 16S rRNA is 99.7% not 100%. We are sorry for this mistake.

Line 96: the probability analysis suggests the relative configuration

It was changed.

Line 100: what does absolutely conserved mean in this context?

The expression was modified. We want to express that the genes related to this angolosamine ring are highly conservative.

Line 162ff: It would be worthwhile to briefly describe the proposed model for medermycin biosynthesis here, especially as key intermediates play a pivotal role for this story.

Thank you for your suggestion. The proposed model for medermycin biosynthesis was added in Figure 2, and it was also described in the main text.

Line 184: reaction conditions were not optimized since the initial conditions were the best, as the authors state in the sentence above.

Thanks for your valuable suggestion. We changed this sentence.

Line 187: why do the authors think, they were not able to identify the important key intermediate?

We carefully examined the LC-MS of all the fractions, but no target compound was found. This was stated in the main text.

Line 189: What is *Euphausia superba*? How was that strain obtained? And why was this strain selected?

Euphausia superba is a marine algae. We isolated the microorganisms from this algae sample. This strain (*Streptomyces* sp. OUCMDZ-4182) has been reported in our previous paper (ref. 41).

We also studied the secondary metabolites of this strain, and (*S*)-DNPA (**10**) had been obtained. We used compound **10** isolated from *Streptomyces* sp. OUCMDZ-4182 as a substrate for the synthesis in our current study.

Line 193: proposed biosynthetic pathways

It was changed.

Line 226: this sentence is highly speculative. Why are proteins the only likely target?

Thank you very much for pointing out this problem for us. We considered this sentence very carefully, and thought it was not appropriate here. There is no basis for us to speculate on the mechanism of antimicrobial action. So, we deleted this speculative statement.

Line 241 ff: The sentence needs to be rephrased as it is difficult to understand in its current form.

Thanks for your comments. This sentence was rephrased.

Line 242ff: ... I am not entirely sure what the authors mean with "which provides new ideas for the efficient application of simple natural products". What does application mean here?

We rewrote this sentence and expressed that this synthesis strategy will use simple and common natural products to form complex skeletons for drug development.

The methods section is incomplete e.g. no methods on how the bioactivity assays were performed are provided

All the experimental methods were provided in the supplementary information.

Raw data should not be made available at "reasonable request" but should be deposited in the appropriate natural product databases (NMR and MS databases)

Source data was provided with this paper.

Fig 3c. What kind of information can be extracted from this panel since the experimental data differ significantly from the calculated data?

We provided some explanations in the figure legend to make it easier to understand. The calculated ECD curves for four plausible stereoisomers are almost same, which suggests that the configurations of sugar unit and the 6-deoxy-dihydrokalafungin (DDHK) moiety have little effect on the ECD Cotton effects. The deciding factor is the stereochemistry at the junction of the two fragments (medermycin and DDHK), which can be determined by these ECD calculation results. So, the key role of the ECD calculations is to determine the stereochemistry at the junction of the two fragments.

Reviewer #2 (Remarks to the Author):

The manuscript 'Chimeric Natural Products Derived from Medermycin and Nature-Inspired Construction of Their Polycyclic Skeletons' is about eight new chimeric medermycin-type natural products with unusual polycyclic skeletons, and some analogues was synthesized, which the biological activity was also studied and some compound presented good activity. The skeleton of the compounds presented in the manuscript was never reported and showed more active than the

positive control, which showed the compound is not only significant for organic synthesis, but also the potential effects in the drug development. The synthetic process developed will offer an effective method for the development for the future.

We are very grateful to you for the support and recognition.

Reviewer #3 (Remarks to the Author):

The accelerating emergence of bacterial resistance to current antibiotics, increasing frequencies of multi-resistant and pan-resistant bacterial pathogens (especially superbugs) and the associated failure of antibiotic treatment in bacterial infections represent the most important problems in today's healthcare. This fact consequently results in an urgent need for novel effective antimicrobial compounds. For the above reasons, I consider the submitted manuscript to be very current.

Unfortunately, I cannot recommend the manuscript in its current form for acceptance. Important data and tests are missing in the microbiological part and these need to be supplemented.

Thank you so much for your constructive and valuable comments and suggestions. We tried our best to improve the manuscript according to these.

Major comments:

1) The methodology for testing antimicrobial activity is incorrectly stated in the supplementary material. It is stated in the text "Pseudomonas aeruginosa ATCC10145, Escherichia coli ATCC11775, Staphylococcus aureus ATCC6538, methicillin-resistant Staphylococcus aureus subsp. aureus ATCC 43300, methicillin-resistant Staphylococcus epidermidis ATCC 35984, Candida albicans ATCC 10231, Candida glabrata ATCC 2001 were evaluated by an agar dilution method." However, the agar diffusion method is described. The results of this method are not given. How the diffusibility of the test substances in the LB and YPD agars used for detection of inhibition zones was determined? What controls were used for this method?

Thanks a lot for your valuable comments and suggestions. We used the agar diffusion method for the preliminary antimicrobial assay, not the agar dilution method. We are very sorry for this mistake. We revised it accordingly. In this assay, zones of inhibition were observed. If a bacteriostatic zone was observed around the paper disk, we will further test the MICs of the active compounds using the broth microdilution method. We did not determine the diffusibility of the compounds, since we used this method to preliminarily determine whether compounds have antimicrobial activity. The quantitative evaluation of antimicrobial activity was done through the broth microdilution method. In the agar diffusion assay, ciprofloxacin lactate and ketoconazole were used as positive controls for bacteria and fungi, respectively.

2) The manufacturer's specification of LB and YPD agars used to determine antimicrobial activity by disk diffusion method is missing.

This information was added in the supplementary information.

3) Although the text suggests a possible reason for the bacterial activity of the test substances (it is stated in the text "Compounds 9 and 26 possess high reactivity as Michael acceptors, probably

resulting in a rapid reaction with free thiols and amines of some important proteins”), I would expect a more detailed description of the possible mechanism of antimicrobial action. Why was ciprofloxacin chosen as a positive control for antibacterial activity? It is on the basis of the presumed mechanism of action that a control antibiotic with a similar type of effect should be chosen.

Thank you very much for your constructive comments. We considered this sentence very carefully, and thought it was not appropriate here. There is no basis for us to speculate on the mechanism of antimicrobial action. So, we deleted this speculative statement. We retested the antibacterial activity, and ciprofloxacin and vancomycin were chosen as positive controls. These two antibiotics can significantly inhibit the methicillin-resistant *Staphylococcus* species and have different mechanisms of antibacterial action.

4) Why are not given all results of antimicrobial activity, ie also MIC results for tested Gram-negative bacteria and yeasts (*Candida albicans* and *Candida glabrata*). I consider it necessary to supplement these results.

The compounds did not show activity against the tested Gram-negative bacteria and yeasts at the maximum concentration what we used (64 $\mu\text{g}/\text{mL}$). These results were added in Table 1.

5) I do not consider the testing of antimicrobial activity to be sufficient. Why the type of effect, ie bactericidal or bacteriostatic, has not been determined. Furthermore, the evaluation of bactericidal effect of tested compounds in time (kill-time assay) and determination of antimicrobial activity in the presence of 4% BSA are missing. I consider it necessary to supplement these tests and results.

Thanks very much for your valuable comments and suggestions. In order to determine that the type of antibacterial effect is bactericidal or bacteriostatic, we tested the MBCs. The time-dependent killing assays on MRSA were also conducted. In addition, we tested the antibacterial activity in the presence of 4% BSA. All these results were showed in Table 1 and Figure 7.

Minor comments

6) The name *Candida glabrata* is wrong, the correct name is *Candida glabrata*.

It was revised.

7) The term *Streptomyces* strains cannot be used, the correct term is *Streptomyces* species strains, resp. *Streptomyces* sp. strains.

Thank you very much for pointing out this mistake. We revised throughout the manuscript including the supplementary information.

Reviewer #4 (Remarks to the Author):

This well-written manuscript by Zhu, Fu and co-workers concerns a series of eight unusual polycyclic marine (mangrove) bacterial products, the chimermycins A–H. These compounds were identified and characterized through a bioguided approach in the search of novel antibiotic agents against drug-resistant bacteria such as MRSA and MRSE. These compounds are chimeric polyketides in part derived from the known pyranonaphthoquinonic medermycin, chimeric natural

products being elaborated from chemical reactions, biocatalyzed or not, between at least two different secondary metabolites.

Besides well-documented descriptions of the structural characterization (including the absolute configuration for several) of these compounds, in particular for the “parent” of this compound series, chimermycin A (1) by full NMR analyses, which were rigorously completed by theoretical calculations, comparisons with closely related natural derivatives, electronic circular dichroism (Cotton effects), hemisynthesis and targeted chemical transformations, the authors identified mild and bioinspired conditions to develop hemisynthesis of chimermycins A, F and H from medermycin in moderate to good yields, as well as a series of analogues.

Comments, questions and suggested corrections:

Fig. 3: I am not totally convinced by the attribution of the absolute stereochemistry of the chimermycin A based on the comparison of the experimental (1) and calculated (1A-D) ECD spectra. Could the authors give additional explanations on their interpretation of these data ?

Thanks a lot for your comments. We provided some explanations in the figure legend to make it easier to understand. The calculated ECD curves for four plausible stereoisomers are almost same, which suggests that the configurations of sugar unit and the 6-deoxy-dihydrokalafungin (DDHK) moiety have little effect on the ECD Cotton effects. The deciding factor is the stereochemistry at the junction of the two fragments (medermycin and DDHK), which can be determined by these ECD calculation results. So, the key role of the ECD calculations is to determine the stereochemistry at the junction of the two fragments.

Page 8: Therefore, the planar part of the structure of chimermycin H (8)...

It was revised.

DNPA: please give full name !

The full name of DNPA was provided in the main text.

Page 8: ketol in place of aldol !? ketone enolate reacting with a ketone, no aldehyde involved... ketolisation ?

It was revised.

Fig. 4 & pages 8 and 10: How was the 4-hydroxyphenylpyruvic acid (13) oxydized into the methylene quinone ?

4-Hydroxyphenylpyruvic acid (13) can be oxydized into quinone methide due to molecular oxygen. The formation of conjugated system promotes this reaction. This was mentioned in the main text.

And also Fig. 4: formation of e ! Dehydrogenation of C-C linkage and reduction of central quinone ? Please explain !

We think that this is a dehydro-aromatization process involving molecular oxygen. This was explained in the manuscript.

And from e, after hemiketalization, how is the decarboxylation promoted to generate the lactone moiety of 8 ?

We think that this is an oxidative decarboxylation. In the process, the hydroxyl group is oxidized to carbonyl group, meanwhile, the carboxyl group is removed. It was also added in the manuscript.

Page 10, second paragraph, line 4: “these conditions”

It was revised.

Page 10: the statement “...the construction of natural product-like compound libraries from medermycin or other polyketides...” is perhaps a bit too preemtuuous, since only half a dozen compounds were made, and only one with another polyketide, granaticin A.

Thanks very much for this comment. We revised this sentence.

Page 11: Antibacterial activity - This section is to my opinion a bit weak, as the quality of the results is not really put forward in terms of perspective and comparison with other antibiotics. Nothing is said about the control used, ciprofloxacin.

We did additional experiments on antibacterial activity. In order to determine that the type of antibacterial effect is bactericidal or bacteriostatic, we tested the MBCs. To further evaluate the effects of three new synthetic compounds (**23**, **24**, and **27**) on bacterial growth, the time-dependent killing assays on MRSA were conducted. We also tested the antibacterial activity in the presence of 4% BSA. All these results were showed in Table 1 and Figure 7. We retested the antibacterial activity, and ciprofloxacin and vancomycin were chosen as positive controls. These two antibiotics can significantly inhibit the methicillin-resistant *Staphylococcus* species and have different mechanisms of antibacterial action.

The following statement: “Compounds 9 and 26 possess high reactivity as Michael acceptors, probably resulting in rapid reaction with free thiols and amines of some important proteins. Their antibacterial activity as well as toxicity may primarily result from this reason” sounds speculative, even if plausible chemistry wise, at least in part. Moreover, other compounds from the tested series, although not active, are also Michael acceptors, so hypotheses are hard to propose on this chemistry feature...

Thank you very much for pointing out this problem for us. We considered this sentence very carefully, and thought it was not appropriate here. There is no basis for us to speculate on the mechanism of antibacterial action. So, we deleted this speculative statement.

SI: enormous and high quality work done on synthesis and characterization. Proton and carbon NMR spectra are of very good to good quality, however, 2D NMR spectra without any annotation of at least diagnostic correlations is not very useful for the reader.

The key annotations were added in the 2D NMR spectra.

In conclusion, this work conveys a very interesting and well-documented piece of chemical science on unusual chimeric marine products. The quantity and quality of the data related to the organic synthesis and the structural characterization work are remarkable. Access to natural product-like compound libraries based on the chemistry of these compounds is just here described by some preliminary examples, nonetheless convincing for the future. However, the work on antibacterial activity is weakly presented. For sure, the best part of this manuscript resides in the bioinspired (and simple but efficient) organic synthesis that has been performed on these complex

natural products and its possible extension to numerous analogues.

Whether or not this manuscript warrants publication in Nature Communications remains an open question at this stage, but I hope that my comments will help the authors to improve their manuscript in order to meet this objective.

Thank you very much for the support and suggestions. We supplemented some experiments on the antibacterial activity and tried our best to improve the manuscript.

REVIEWER COMMENTS

Reviewer #1 (Remarks to the Author):

The manuscript “Chimeric natural products derived from medermycin and natural-inspired construction of their polycyclic skeletons” by Fu and co-workers has been significantly improved. Most of my comments on the paper have been addressed. Since I suggested such major rephrasing and reorganization of the main text, the authors rewrote a substantial part of the paper. During this process, a couple of things were added that I believe the authors should revise. I believe that the language of the paper needs to be improved both in terms of grammar, style, and the usage of the correct words and tense. For some reason the introduction now contains results (MS network analysis). Please remove any results from the introduction! Throughout the manuscript instruments were named instead of the data that were acquired using these instruments (e.g. line 82). Moreover, the Methods section is now split into two subsections that are separated by most of the SI information. New experiments were added, yet the corresponding methods section (genome sequencing and genome mining) is missing. I do not understand the logic of the biosynthesis section where first the metabolic level is described before the gene architecture is discussed (the corresponding Figure displays gene cluster and biosynthesis in the more intuitive order). Only the biosynthesis of medermycin is discussed. What about the other fragments that are required for the formation of the chimeric natural products described in the manuscript? The authors state that the biosynthetic machinery seems to be virtually the same as in the reported producers of medermycins. Can the authors speculate why no chimeric natural products can be detected in the other producers if it is a spontaneous reaction? I believe that a lot of information has been added to the antibacterial activity section that can be significantly streamlined. The authors have followed my suggestion to put their study in a broader scientific context. Unfortunately, some crucial explanations are missing though. The authors mention “progress” in the field but do not specify what this progress is, the author propose “more rational research methods” but fail to explain what these methods would be. Moreover, the authors hypothesize that “analysis of biosynthetic genes would help them discover key nonenzymatic steps”. It is not clear to me how transformations that are spontaneous and do not require enzymes and hence no genes that encode these enzymes can be identified by analyzing genes whose products are not involved in these transformations.

Abstract:

Line 18: medermycins are not a class of antibiotics. They are rather a family.

Line 19: “The PK biosynthetic pathways of this kind of NP is stable and clear”. Rephrase “this kind of NP” and specify what “stable” and “clear” means.

Line 28: rephrase “we offer a valuable idea”.

Introduction:

Line 32: add a reference.

Line 33: change to “mechanism of action”.

Line 36: what is a “target molecule for biosynthesis”?

Line 37: The authors have rephrased the sentence but adding two terms (genome mining, heterologous expression) without further explanation is not sufficient.

Line 42: it is not clear what “among them” is referring to.

Line 49: “could” has an ambiguous meaning. Try to avoid “could” statements throughout the text and change to are formed.

Line 58: “reliable” is not the right word here.

Line 67: the authors should try to annotate the other medermycin analogs since they only differ in methyl groups and other simple changes that can be easily annotated using MS2 data.

Results:

Line 76: I would still prefer a one sentence introduction into the results section to understand what the plan of the authors was with the strain other than producing a lot of extract. All the relevant information for some reason ended up in the introduction. The level of detail is too much for the introduction and the information given in the introduction would be better suited for the beginning of the results section.

Line 94: I think the authors mean to say conserved instead of conservative.

Fig 2: Please remove the percentages and replace it with some sort of color scheme.

Line 130: rephrase “due to molecular oxygen”.

Line 133: “these look like nonenzymatic reactions” sounds colloquial and should be phrased.

Line 136: the gene clusters differ by the presence or absence of two genes. What is their function?

Line 139: How do the authors explain that the gene clusters are, according to the authors, virtually identical, yet not every producer biosynthesizes the spontaneous chimeras? What about the spontaneous reaction partner? The authors never mention the identification of the genes responsible for the biosynthesis of 12.

Line 142: what does “preliminary verified” mean? Please rephrase.

Line 189: "these results gave an inspiration" please rephrase.

Line 210: rephrase the sentence containing "obviously".

Line 215: exchange the word "fixed" with a more appropriate word.

Line 233: rephrase sentence containing "will realize".

Line 244: what do the authors mean with some "progress in nonenzymatic biosynthesis" was made?

Line 246: What do the authors mean with "explore more rational research methods"? What methods would these be?

Line 249: What do the authors mean with "analysis of biosynthetic genes can also help us to discover key nonenzymatic steps"? This seems contradictory to me since nonenzymatic reactions do not require enzymes and hence no biosynthetic genes involved in spontaneous reactions exist.

Reviewer #3 (Remarks to the Author):

First of all, I would like to thank the authors for editing the manuscript based on my previous comments. I believe that the text is now more accurate.

However, I still have the following three comments:

1) In my opinion it is not necessary to repeat the explanation of abbreviations in the text, see MRSA on lines 63 + 198 and MRSE on lines 74 + 198.

2) I recommend modifying the sentence „Given the antibiotic resistance of MRSA and MRSE, ciprofloxacin and vancomycin were chosen as positive controls for bacteria“ to „Given the antibiotic resistance of MRSA and MRSE, ciprofloxacin and vancomycin were chosen as positive controls of antibacterial activity“.

3) I cannot agree with the authors that compounds 9, 23, and 24 may be bacteriostatic agents. The fact is that the MBC values are higher, but this does not mean that this is a bacteriostatic effect. The obtained MBC values prove that, although in a higher concentrations, there was an killing of the tested bacterial strains. The bacteriostatic effect can be observed in the case of compound number 1. I consider it necessary to remove the information on the bacteriostatic effect of the tested compounds from the text.

Reviewer #5

< Editor's note: In comments to the Editorial office, the reviewer expressed that the concerns from the reviewers in the previous round were sufficiently met. >

Response to referees

We thank the reviewers for their constructive comments and suggestions, which helped improve the quality of our manuscript. According to these comments and suggestions, we have revised the manuscript very carefully. The changes are highlighted in a **yellow background**. Comments from separate reviewers are listed below and our replies are in **blue**.

Reviewer #1 (Remarks to the Author):

The manuscript “Chimeric natural products derived from medermycin and natural-inspired construction of their polycyclic skeletons” by Fu and co-workers has been significantly improved. Most of my comments on the paper have been addressed. Since I suggested such major rephrasing and reorganization of the main text, the authors rewrote a substantial part of the paper. During this process, a couple of things were added that I believe the authors should revise. I believe that the language of the paper needs to be improved both in terms of grammar, style, and the usage of the correct words and tense.

Thank you very much for the support and suggestions. The language of the manuscript has been polished and corrected by Springer Nature Author Services.

For some reason the introduction now contains results (MS network analysis). Please remove any results from the introduction!

The results have been removed from the introduction.

Throughout the manuscript instruments were named instead of the data that were acquired using these instruments (e.g. line 82).

The data has been added and the similar problems have been revised.

Moreover, the Methods section is now split into two subsections that are separated by most of the SI information. New experiments were added, yet the corresponding methods section (genome sequencing and genome mining) is missing.

Some methods have been moved to the main text from the SI. Meanwhile, the methods of genome sequencing and analysis have been added.

I do not understand the logic of the biosynthesis section where first the metabolic level is described before the gene architecture is discussed (the corresponding Figure displays gene cluster and biosynthesis in the more intuitive order).

Thank you very much for your comments. According to the suggestions, the order of the biosynthesis section has been adjusted.

Only the biosynthesis of medermycin is discussed. What about the other fragments that are required for the formation of the chimeric natural products described in the manuscript?

Thanks for your valuable comments. We investigated some literatures about the formation mechanism of butane-2,3-dione (**12**) and 4-hydroxyphenylpyruvic acid (**13**). The references and the explanations have been added in the manuscript.

The authors state that the biosynthetic machinery seems to be virtually the same as in the reported producers of medermycins. Can the authors speculate why no chimeric natural products can be detected in the other producers if it is a spontaneous reaction?

We thought the reason why other producers of medermycin do not produce the spontaneous chimeras may be the difference of fermentation conditions. In our study, we used the rice-based medium for the fermentation with a relatively long growth time (60 days), while most of the reported studies on the secondary metabolites of other medermycin producers used liquid medium in the shaker. We did a small fermentation using liquid medium for the strain OUCMDZ-4982, and no chimeras were observed. So, we believe that the fermentation condition led to the formation of these chimeric compounds.

I believe that a lot of information has been added to the antibacterial activity section that can be significantly streamlined. The authors have followed my suggestion to put their study in a broader scientific context.

Thank you very much for your approval.

Unfortunately, some crucial explanations are missing though. The authors mention “progress” in the field but do not specify what this progress is, the author propose “more rational research methods” but fail to explain what these methods would be. Moreover, the authors hypothesize that “analysis of biosynthetic genes would help them discover key nonenzymatic steps”. It is not clear to me how transformations that are spontaneous and do not require enzymes and hence no genes that encode these enzymes can be identified by analyzing genes whose products are not involved in these transformations.

Thanks for your valuable suggestions and comments. We have added some explanations and modified the corresponding expressions for these issues. The point-by-point response has been shown in the corresponding places below.

Abstract:

Line 18: medermycins are not a class of antibiotics. They are rather a family.

It has been changed.

Line 19: “The PK biosynthetic pathways of this kind of NP is stable and clear”. Rephrase “this kind of NP” and specific what “stable” and “clear” means.

This sentence has been rewritten. The expression has been modified to make it easier to understand.

Line 28: rephrase “we offer a valuable idea”.

It has been revised.

Introduction:

Line 32: add a reference.

The appropriate reference has been added.

Line 33: change to “mechanism of action”.

It has been changed.

Line 36: what is a “target molecule for biosynthesis”?

We have reorganized this sentence.

Line 37: The authors have rephrased the sentence but adding two terms (genome mining, heterologous expression) without further explanation is not sufficient.

These two terms have been explained in more detail.

Line 42: it is not clear what “among them” is referring to.

We have rewritten this sentence and deleted “among them”.

Line 49: “could” has an ambiguous meaning. Try to avoid “could” statements throughout the text and change to are formed.

It has been changed and other similar problems throughout the text have also been revised.

Line 58: “reliable” is not the right word here.

It has been changed to “important”.

Line 67: the authors should try to annotate the other medermycin analogs since they only differ in methyl groups and other simple changes that can be easily annotated using MS2 data.

Some other medermycin analogs have been annotated in Supplementary Fig. 1. However, chimedermycins B–D (2–4) cannot be found in the molecular networking. Therefore, we speculated that the methyl ester derivatives of chimedermycin A (1) may be formed in the process of separation and purification in the presence of methanol.

Results:

Line 76: I would still prefer a one sentence introduction into the results section to understand what the plan of the authors was with the strain other than producing a lot of extract. All the relevant information for some reason ended up in the introduction. The level of detail is too much for the introduction and the information given in the introduction would be better suited for the beginning of the results section.

Thanks for your valuable comments. We have revised this section to further clarify our plan to study this strain. Some detailed information has been moved to the results section from the introduction.

Line 94: I think the authors mean to say conserved instead of conservative.

Thanks for your valuable suggestion. “conservative” has been changed to “conserved”.

Fig 2: Please remove the percentages and replace it with some sort of color scheme.

The percentages have been removed.

Line 130: rephrase “due to molecular oxygen”.

This sentence has been revised.

Line 133: “these look like nonenzymatic reactions” sounds colloquial and should be phrased.

This expression has been revised.

Line 136: the gene clusters differ by the presence or absence of two genes. What is their function?

The function of these two genes has been explained in the text.

Line 139: How do the authors explain that the gene clusters are, according to the authors, virtually identical, yet not every producer biosynthesizes the spontaneous chimeras? What about the spontaneous reaction partner? The authors never mention the identification of the genes responsible for the biosynthesis of 12.

Thanks a lot for your constructive comments on this important issue. We thought the reason why other producers of medermycin do not produce the spontaneous chimeras may be the difference of fermentation conditions. In our study, we used the rice-based medium for the fermentation with a relatively long growth time (60 days), while most of the reported studies on the secondary metabolites of other medermycin producers used liquid medium in the shaker. We did a small fermentation using liquid medium for the strain OUCMDZ-4982, and no chimeras were observed. So, we believe that the fermentation condition led to the formation of these chimeric compounds. We investigated some literatures about the formation mechanism of butane-2,3-dione (**12**) and found that this compound can be formed through some different pathways. It can be formed from sugars and lipids via microbial metabolism or certain chemical reactions, such as lipid oxidation, carbohydrate decomposition, and the Maillard reaction. So, this compound may be formed from the media components during the fermentation process. The related references have been added.

Line 142: what does “preliminary verified” mean? Please rephrase.

It has been revised.

Line 189: “these results gave an inspiration” please rephrase.

It has been revised.

Line 210: rephrase the sentence containing “obviously”.

This sentence has been rewritten.

Line 215: exchange the word “fixed” with a more appropriate word.

It has been changed to “stable”, and this sentence has been rewritten.

Line 233: rephrase sentence containing “will realize”.

This sentence has been rewritten.

Line 244: what do the authors mean with some “progress in nonenzymatic biosynthesis” was made?

This sentence has been reorganized. We were trying to express that the nonenzymatic formation mechanism of some natural products has been elucidated, but there are still some challenges in this process.

Line 246: What do the authors mean with “explore more rational research methods”? What methods would these be?

Thanks a lot for your constructive comment on this issue. It has been revised. Small molecule probes with chemical activity could probably be used to explore the nonenzymatic formation of natural products.

Line 249: What do the authors mean with “analysis of biosynthetic genes can also help us to discover key nonenzymatic steps”? This seems contradictory to me since nonenzymatic reactions do not require enzymes and hence no biosynthetic genes involved in spontaneous reactions exist.

Thanks for your valuable comment. To avoid misunderstanding, we have rewritten this sentence and expressed that the analysis of biosynthetic genes can help us speculate whether these processes are nonenzymatic.

Reviewer #3 (Remarks to the Author):

First of all, I would like to thank the authors for editing the manuscript based on my previous comments. I believe that the text is now more accurate.

We are very grateful to you for the support.

However, I still have the following three comments:

1) In my opinion it is not necessary to repeat the explanation of abbreviations in the text, see MRSA on lines 63 + 198 and MRSE on lines 74 + 198.

It has been revised.

2) I recommend modifying the sentence “Given the antibiotic resistance of MRSA and MRSE, ciprofloxacin and vancomycin were chosen as positive controls for bacteria” to “Given the antibiotic resistance of MRSA and MRSE, ciprofloxacin and vancomycin were chosen as positive controls of antibacterial activity”.

We have modified this sentence.

3) I cannot agree with the authors that compounds 9, 23, and 24 may be bacteriostatic agents. The fact is that the MBC values are higher, but this does not mean that this is a bacteriostatic effect. The obtained MBC values prove that, although in a higher concentration, there was a killing of the tested bacterial strains. The bacteriostatic effect can be observed in the case of compound number 1. I consider it necessary to remove the information on the bacteriostatic effect of the tested compounds from the text.

Thanks for your comments. This information has been removed, and the expression involved has been reorganized.

Reviewer #5 (Remarks to the Author):

< Editor's note: In comments to the Editorial office, the reviewer expressed that the concerns from

the reviewers in the previous round were sufficiently met. >

Thank you so much for your support and approval.

REVIEWER COMMENTS

Reviewer #1 (Remarks to the Author):

Most of my comments were addressed and as a result the manuscript significantly improved. Two or three of my previous comments should be addressed in more detail (e.g. biosynthesis of butanedione). If the comments listed below are included in a final round of very minor revisions, I am highly supportive of publishing the manuscript in Nat. Commun. Some things, and I am not sure here if this was done by the Springer Nature Author Services or the authors, have been changed and are now incorrect. Since these changes were not part of any of the comments made by any of the reviewers, I suspect that they were introduced by accident during the language polishing process by Springer Nature Author Services. These changes include for example changing Gram-positive into gram-positive. Gram is the name of the person who invented the stain and hence Gram needs to remain capitalized.

In line 20: "the biosynthesis of this family of natural products is conserved in bacteria" suggests that all bacteria are capable of producing these compounds which is not true. `

Line 28 I think the sentence "This work offers a way to understand..." can be rephrased to make it sound a little better.

Line 37: The authors state in their response to the reviewer's comments that terms like "genome mining" "have now been explained in more detail". This, however, is not the case. Please add at least a sentence on how genomic information aids in the discovery of novel natural product scaffolds.

Line 50: The sentence: "Discoipyrrole A is formed from three starting secondary metabolites" is wrong. Anthranilic acid for example is a primary metabolite that is a key intermediate in tryptophan biosynthesis.

Line 56: The sentence: "which makes the expansion of natural product diversity related to the nonenzymatic reactions more significant" is a little awkward. Please rephrase.

Line 88: "DDHK moiety was confirmed by the heteronuclear multiple bond correlation (HMBC) correlations...". Please rephrase to avoid having the same noun right next to each other.

Line 122: what is a gene whose gene product is involved in "activating acyl carrier proteins"? A Pptase?

Line 148: My question on the origin on the other components of the spontaneous reaction such as butanedione was aiming at the enzymes involved in the biosynthesis of the compound rather than the pathways that provide the precursors such as pyruvate. Based on my very brief research, butanedione is not produced by every bacterium. As a result, the presence or absence of the genes whose gene products are involved in butanedione biosynthesis might explain why the chimeric products can only be found in some species. There is for example a KEGG pathway that shows how butanedione is biosynthesized. Please exchange the generic information on which general cellular processes lead to precursors of butanedione with how butanedione is biosynthesized from ubiquitous primary metabolites and whether the corresponding genes are present in the producers of medermycins.

Line 249: How would small molecule probes help in identifying spontaneous reactions? These kind of probes have been used for quite some time to identify natural products which have highly reactive functional groups. How would you be able to distinguish between reactive functional groups as warheads and spontaneous reactions towards chimeric natural products?

Response to referees

We thank the reviewer for the constructive comments and suggestions, which helped improve the quality of our manuscript. According to these comments and suggestions, we have revised the manuscript very carefully. The changes are highlighted in a **yellow background**. Comments from the reviewer are listed below and our replies are in **blue**.

Reviewer #1 (Remarks to the Author):

Most of my comments were addressed and as a result the manuscript significantly improved. Two or three of my previous comments should be addressed in more detail (e.g. biosynthesis of butanedione). If the comments listed below are included in a final round of very minor revisions, I am highly supportive of publishing the manuscript in Nat. Commun. Some things, and I am not sure here if this was done by the Springer Nature Author Services or the authors, have been changed and are now incorrect. Since these changes were not part of any of the comments made by any of the reviewers, I suspect that they were introduced by accident during the language polishing process by Springer Nature Author Services. These changes include for example changing Gram-positive into gram-positive. Gram is the name of the person who invented the stain and hence Gram needs to remain capitalized.

Thank you very much for the comments and suggestions. We have double checked the manuscript and revised the incorrect changes.

In line 20: “the biosynthesis of this family of natural products is conserved in bacteria” suggests that all bacteria are capable of producing these compounds which is not true.

This sentence has been revised to “The biosynthesis of this family of natural products has been studied clearly, and new skeletons related to medermycin have rarely been reported until recently”.

Line 28: I think the sentence “This work offers a way to understand...” can be rephrased to make it sound a little better.

We have revised this sentence to “This work paves the way for understanding the nonenzymatic formation of complex natural products and using it to synthesize natural product derivatives”.

Line 37: The authors state in their response to the reviewer’s comments that terms like “genome mining” “have now been explained in more detail”. This, however, is not the case. Please add at least a sentence on how genomic information aids in the discovery of novel natural product scaffolds.

Thanks very much for your valuable comment. We have revised this part and added two sentences to explain how genomic information aids in the discovery of novel natural product scaffolds. “Researchers can use genomic sequence data to identify and predict genes that encode the production of novel compounds. Furthermore, genomic information can guide scientists to activate cryptic gene clusters through genetic manipulations to obtain new natural products”.

Line 50: The sentence: “Discoipyrrole A is formed from three starting secondary metabolites” is wrong. Anthranilic acid for example is a primary metabolite that is a key intermediate in

tryptophan biosynthesis.

This sentence has been revised to “Discoipyrrole A is formed from three starting metabolites”.

Line 56: The sentence: “which makes the expansion of natural product diversity related to the nonenzymatic reactions more significant” is a little awkward. Please rephrase.

To avoid misunderstanding, this sentence has been revised to “In addition to their fascinating structures, these compounds formed through nonenzymatic reactions also exhibit potent biological activities”.

Line 88: “DDHK moiety was confirmed by the heteronuclear multiple bond correlation (HMBC) correlations...”. Please rephrase to avoid having the same noun right next to each other.

It has been revised.

Line 122: what is a gene whose gene product is involved in “activating acyl carrier proteins”? A Pptase?

Thank you very much. According to your comment and the reference, we revised this sentence to “The missing gene *med-21* was thought to be associated with regulation function (a possible kinase gene), and the other one *med-24* was proposed to encode a phosphopantetheinyl transferase”.

Line 148: My question on the origin on the other components of the spontaneous reaction such as butanedione was aiming at the enzymes involved in the biosynthesis of the compound rather than the pathways that provide the precursors such as pyruvate. Based on my very brief research, butanedione is not produced by every bacterium. As a result, the presence or absence of the genes whose gene products are involved in butanedione biosynthesis might explain why the chimeric products can only be found in some species. There is for example a KEGG pathway that shows how butanedione is biosynthesized. Please exchange the generic information on which general cellular processes lead to precursors of butanedione with how butanedione is biosynthesized from ubiquitous primary metabolites and whether the corresponding genes are present in the producers of medermycins.

Thank you very much. According to your comments, we revised this section. The biosynthetic pathway of butanedione in bacteria has been investigated and added to the manuscript. We also searched the genes related to the biosynthesis of butanedione in the genome of *Streptomyces* sp. OUCMDZ-4982. This section has been revised to “Butane-2,3-dione may be produced by *Streptomyces* sp. OUCMDZ-4982. It has been reported that various bacteria can produce butane-2,3-dione. In all these producers, acetohydroxyacid synthase (AHAS; acetolactate synthase) catalyzes pyruvate to generate acetolactate, an important intermediate of the biosynthesis of branched-chain amino-acids. Acetolactate is unstable and can be easily converted to butane-2,3-dione through nonenzymatic decarboxylation in the presence of oxygen.⁴² AHAS is a tetramer composed of two kinds of subunits, a catalytic one and a regulatory one, which are encoded by the genes *ilvB* (GenBank accession no. AY785370.1) and *ilvN* (GenBank accession no. AF175526.1), respectively, in a *Streptomyces cinnamonensis* strain.^{43,44} We searched these genes in the genome of *Streptomyces* sp. OUCMDZ-4982, and found two genes with high homology to *ilvB* (91% identity) and *ilvN* (90% identity). So, we speculated that *Streptomyces* sp. OUCMDZ-4982 has the ability to produce butane-2,3-dione.”

Line 249: How would small molecule probes help in identifying spontaneous reactions? These kind of probes have been used for quite some time to identify natural products which have highly reactive functional groups. How would you be able to distinguish between reactive functional groups as warheads and spontaneous reactions towards chimeric natural products?

Thanks very much for your constructive questions. Like you said, some probes can be used to identify natural products with highly reactive functional groups or structural fragments. Some of these reactive units can react with other compounds to form chimeric skeletons through spontaneous reactions under suitable conditions. Such spontaneous reactions do not occur on all the natural products with reactive units, but the high chemical reactivity is the basis of spontaneous reactions. So, the probes can help us identify natural products with high chemical reactivity. Further, we may find some spontaneous reactions towards chimeric natural products from these reactive compounds. Thus, we have revised the sentence to “In the future, we need to explore more rational research methods to further understand nonenzymatic biosynthesis, such as utilizing small molecule probes with chemical activity to explore the natural products with highly reactive functional groups or structural fragments. These compounds may have the potential to form chimeric skeletons through spontaneous reactions”.

REVIEWERS' COMMENTS

Reviewer #1 (Remarks to the Author):

Abstract:

Line 20: please remove the word “clearly”.

Introduction:

Line 38: Change sentence to: “Researchers use genomic sequence information to identify and annotate biosynthetic gene clusters whose gene products are involved in the biosynthesis of novel natural product scaffolds”. The next sentence can be removed as it is not related to the research presented in the manuscript and it is unrelated to genome mining.

Main text:

Line 122: Change to: “The product of the missing gene med-21 is likely associated with pathway regulation (putative kinase) and med-24 encodes a putative phosphopantetheinyl transferase”

Line 149 ff: According to the KEGG pathway AHAS is only one of several enzymes that are required to convert pyruvate to butane-dione. A key intermediate seems to be acetoin.

Line 151: Something about the sentence “...catalyzes pyruvate to generate acetolactate” seems to be not right. Please rephrase. An enzyme cannot catalyze pyruvate.

Line 154: change to “AHAS is a heterotetramer that is composed of two subunits...”

Line 159: My question was, are these genes present in *Streptomyces* sp. AM-7161 which seems not to be capable of producing the compounds that require butanedione as a building block. Does this explain why one strain produces the chimeras and the other one does not?

Line 258ff: remove “the” before natural products. The problem with the following sentence is that every natural product that forms covalent bonds with its target molecule will be picked up with the

probe strategy and the number of molecules that will form natural product chimeras among these is likely very very low. I am not sure if this strategy will really lead to the identification of spontaneous reactions. As mentioned in my previous comments, this probe-based approach has been followed by several groups in the field to identify natural products with a certain type of warhead. There are groups out there whose entire work is based on the assumption that you can synthesize probes to fish out natural products with a certain reactivity. Maybe consider adding references where appropriate (e.g. utilizing small molecule probes with chemical activity to explore natural products with highly reactive functional groups) to indicate that this idea is not new.

Response to referees

We thank the reviewer and editor for their constructive comments and suggestions, which helped improve the quality of our manuscript. According to these comments and suggestions, we have revised the manuscript very carefully. The changes are highlighted in a yellow background. Comments from the reviewer are listed below and our replies are in blue.

Reviewer #1 (Remarks to the Author):

Abstract:

Line 20: please remove the word “clearly”.

It has been removed.

Introduction:

Line 38: Change sentence to: “Researchers use genomic sequence information to identify and annotate biosynthetic gene clusters whose gene products are involved in the biosynthesis of novel natural product scaffolds”. The next sentence can be removed as it is not related to the research presented in the manuscript and it is unrelated to genome mining.

It has been changed and its next sentence has been removed.

Main text:

Line 122: Change to: “The product of the missing gene med-21 is likely associated with pathway regulation (putative kinase) and med-24 encodes a putative phosphopantetheinyl transferase”

It has been changed.

Line 149 ff: According to the KEGG pathway AHAS is only one of several enzymes that are required to convert pyruvate to butane-dione. A key intermediate seems to be acetoin.

The expression has been modified. According to the results reported in the literatures, we think the intermediate is acetolactate. This compound is not stable and can be easily converted to butanedione through nonenzymatic decarboxylation in the presence of oxygen.

Line 151: Something about the sentence “...catalyzes pyruvate to generate acetolactate” seems to be not right. Please rephrase. An enzyme cannot catalyze pyruvate.

This sentence has been rephrased.

Line 154: change to “AHAS is a heterotetramer that is composed of two subunits...”

It has been changed.

Line 159: My question was, are these genes present in *Streptomyces* sp. AM-7161 which seems not to be capable of producing the compounds that require butanedione as a building block. Does this explain why one strain produces the chimeras and the other one does not?

Thank you very much. This is a very good suggestion. However, the whole genome sequence of *Streptomyces* sp. AM-7161 is not available. Furthermore, the nonenzymatic formation of the chimeras requires not only precursors but also suitable chemical conditions. The production of these chimeras may be resulted from the composite effect of many factors.

Line 258ff: remove “the” before natural products. The problem with the following sentence is that every natural product that forms covalent bonds with its target molecule will be picked up with the probe strategy and the number of molecules that will form natural product chimeras among these is likely very very low. I am not sure if this strategy will really lead to the identification of spontaneous reactions. As mentioned in my previous comments, this probe-based approach has been followed by several groups in the field to identify natural products with a certain type of warhead. There are groups out there whose entire work is based on the assumption that you can synthesize probes to fish out natural products with a certain reactivity. Maybe consider adding references where appropriate (e.g. utilizing small molecule probes with chemical activity to explore natural products with highly reactive functional groups) to indicate that this idea is not new.

Thanks a lot for your suggestions. We deliberated this question very carefully and revised the corresponding sentence. This strategy really needs to be further improved in our follow-up work. Meanwhile, three related references have been added.